# Targeting BC200/miR218-5p Signaling Axis for Overcoming Temozolomide Resistance and Suppressing Glioma Stemness

**DOI:** 10.3390/cells9081859

**Published:** 2020-08-08

**Authors:** Yu-Kai Su, Jia Wei Lin, Jing-Wen Shih, Hao-Yu Chuang, Iat-Hang Fong, Chi-Tai Yeh, Chien-Min Lin

**Affiliations:** 1Graduate Institute of Clinical Medicine, College of Medicine, Taipei Medical University, Taipei City 11031, Taiwan; yukai.su@gmail.com (Y.-K.S.); ns246@tmu.edu.tw (J.W.L.); ctyeh@s.tmu.edu.tw (C.-T.Y.); 2Department of Neurology, School of Medicine, College of Medicine, Taipei Medical University, Taipei City 11031, Taiwan; impossiblewasnothing@hotmail.com; 3Division of Neurosurgery, Department of Surgery, Taipei Medical University-Shuang Ho Hospital, New Taipei City 23561, Taiwan; 4Taipei Neuroscience Institute, Taipei Medical University, Taipei City 11031, Taiwan; 5Graduate Institute of Cancer Biology and Drug Discovery, College of Medical Science and Technology, Taipei Medical University, Taipei 11031, Taiwan; shihjw@tmu.edu.tw; 6Ph.D. Program for Cancer Biology and Drug Discovery, College of Medical Science and Technology, Taipei Medical University, Taipei 11031, Taiwan; 7Department of Neurosurgery, An Nan Hospital, China Medical University, Tainan 70965, Taiwan; greeberg1975@gmail.com; 8Department of Medical Research & Education, Taipei Medical University-Shuang Ho Hospital, New Taipei City 23561, Taiwan; 9Department of Medical Laboratory Science and Biotechnology, Yuanpei University of Medical Technology, Hsinchu 300, Taiwan

**Keywords:** non-coding RNA, noncoding RNA BC200/miR-218-5p signaling circuit, glioblastoma stem cells, temozolomide resistance

## Abstract

*Background*: Glioblastoma (GB) is one of the most common (~30%) and lethal cancers of the central nervous system. Although new therapies are emerging, chemoresistance to treatment is one of the major challenges in cancer treatment. Brain cytoplasmic 200 (BC200) RNA, also known as BCYRN1, is a long noncoding RNA (lncRNA) that has recently emerged as one of the crucial members of the lncRNA family. BC200 atypical expression is observed in many human cancers. BC200 expression is higher in invasive cancers than in benign tumors. However, the clinical significance of BC200 and its effect on GB multiforme is still unexplored and remains unclear. *Methods*: BC200 expression in GB patients and cell lines were investigated through RT-qPCR, immunoblotting, and immunohistochemistry analysis. The biological importance of BC200 was investigated in vitro and in vivo through knockdown and overexpression. Bioinformatic analysis was performed to determine miRNAs associated with BC200 RNA. *Results*: Our findings revealed that in GB patients, BC200 RNA expression was higher in blood and tumor tissues than in normal tissues. BC200 RNA expression have a statistically significant difference between the IDH1 and P53 status. Moreover, the BC200 RNA expression was higher than both p53, a prognostic marker of glioma, and Ki-67, a reliable indicator of tumor cell proliferation activity. Overexpression and silencing of BC200 RNA both in vitro and in vivo significantly modulated the proliferation, self-renewal, pluripotency, and temozolomide (TMZ) chemo-resistance of GB cells. It was found that the expressions of BC200 were up-regulated and that of miR-218-5p were down-regulated in GB tissues and cells. miR-218-5p inhibited the expression of BC200. *Conclusions*: This study is the first to show that the molecular mechanism of BC200 promotes GB oncogenicity and TMZ resistance through miR-218-5p expression modulation. Thus, the noncoding RNA BC200/miR-218-5p signaling circuit is a potential clinical biomarker or therapeutic target for GB.

## 1. Background

Glioblastoma (GB) is one of the most common (~30%) and lethal cancers of the central nervous system [1]. The World Health Organization classifies glioma into grades I to IV based on its histopathologically determined malignancy level [2]. GB, a grade IV astrocytoma, is highly aggressive, malignant and invasive [3]. Despite improved therapeutic interventions, the treatment of high-grade glioma is challenging, even after combining radiation therapy with surgical resection [4]. Therefore, the molecular mechanism of GB pathogenesis and treatment resistance must be explored. With the advances in sequencing technologies, evidence indicates a regulatory role of long noncoding RNA (lncRNA) [5]. Recent studies have shown that lncRNAs play a role in the development, progression, and metastasis [6] of many cancers, including GB [7]. These lncRNAs consist of >200 nucleotides and cannot translate proteins. Furthermore, studies have reported dysregulated lncRNA expressions, which determine the temozolomide (TMZ)-drug sensitivity or TMZ-drug resistance of GB [8]. This lncRNA exerts its functions through lncRNA–miRNA interactions and mRNA silencing [9].

BCYRN1, also known as brain cytoplasmic 200 (BC200) RNA, is a lncRNA highly expressed in the brain (neuron-specific transcript) in the presence of neurodegenerative diseases [10]. Sometimes, BC200 RNA expression is atypically elevated in the presence of various human cancers [11] and its expression is higher in invasive cancer cells than in benign tumors [12]. BC200 RNA was also found in blood specimens such as breast cancer and hepatocellular carcinoma [13,14]. These data suggest the BC200 RNA has potential value as a diagnostic and prognostic marker for human cancers. However, information regarding BC200 RNA as a clinical biomarker or therapeutic target for GB is still lacking. The exact underlying molecular mechanism of BC200 RNA in glioma development and progression is still under investigation.

In this study, the role of BC200 RNA in GB was evaluated to understand the molecular mechanism of chemoresistance and disease progression. The expression level of BC200 RNA in the tissue of GB patients and GB cell lines was evaluated. The sulforhodamine B (SRB) assay was used to determine the effect of TMZ treatment on GB cells. Furthermore, the stem cell population was evaluated through flow cytometric analysis of GB cell lines. Overexpression and silencing of BC200 RNA showed its role in survival and GB tumorigenesis reduction both in vitro and in vivo. Ours is the first study to show the molecular mechanism of BC200 RNA in the promotion of TMZ resistance in GB through miR-218-5p expression modulation. Thus, BC200 RNA is suggested as a potential clinical biomarker or therapeutic target for GB.

## 2. Materials and Methods

### 2.1. Patients and Tumor Samples

The study was approved by the Joint Institutional Review Board (JIRB) of the Taipei Medical University—Shuang Ho Hospital. Written informed consents were obtained from all participants (Approval number: N201903047). Tissue samples from patients with primary and recurrent GB were obtained from the Taipei Medical University-Shuang Ho Hospital GB cohort. Surgically resected tissues samples were immediately frozen in RNALater (QIAGEN, Hilden, Germany) and subsequently stored at −80 °C until use.

### 2.2. Cell Lines and Cell Culture

Human GB cell line T98G (ATCC^®^ CRL-1690™) (ATCC, Manassas, VA, USA) and U87MG (ATCC^®^ HTB-14™) (ATCC) were cultured in DMEM; DBTRG-05MG (ATCC^®^ CRL-2020™) (ATCC) and GBM8901 (Bioresource Collection and Research Center, Hsinchu, Taiwan) were cultured in RPMI. In order to demonstrate the association between BC200 and MGMT, the U87MG cells were transfected to overexpress MGMT. The human MGMT open reading frame (ORF) plasmid was purchased from OriGene (Cat# MGMT (RC229131, Taipei, Taiwan) and the cells transfected according to vendor’s instructions; MGMT-overexpressing U87MG cells were then allowed to grow at 37 °C in 5% CO_2_ humidified atmosphere in Dulbecco’s modified Eagle’s medium (DMEM). Media was supplemented with 10% fetal bovine serum, and streptomycin (100 μg/mL), penicillin (100 IU/mL), and at 80% confluency the cells were sub-cultured every 2–3 days. The expression of MGMT was verified by western blots.

### 2.3. Sulforhodamine B (SRB) Viability Assay

T98G, U87MG, DBTRG-05MG and GBM8901 cells were seeded in 96-well plates in triplicates at a concentration of 3000 cells per well. After 24 h incubation in a 5% CO_2_ humidified incubator at 37 °C, the cells were treated with varying concentrations of 0–1000 μM TMZ as indicated for 24 h. Thereafter, cells were washed in PBS twice, fixed in cold 10% trichloroacetic acid (TCA) for 1h, washed with distilled water, and then incubated in 0.4 SRB (*w*/*v*) in 1% acetic acid at room temperature for 1 h. After washing of unbound SRB dye with 1% acetic acid thrice, the plates were air-dried. Attached dye was dissolved in 20 mM trizma base, and absorbance was read in a microplate reader at a wavelength of 570 nm. (Molecular Devices, Sunnyvale, CA, USA).

### 2.4. Cell Proliferation Assay

Cell Counting Kit-8 (CCK-8, Dojindo Laboratories, Rockville, MD, USA) was applied for detecting the cell proliferation. T98G, U87MG, DBTRG-05MG and GBM8901 cells were seeded in 96-well plates in triplicates, incubated for 24, 48, 72 or 96 hrs. For each time point, 10 μL CCK-8 solution was added per well and the cells were incubated at 37 °C for 2 h and absorbance was read in a microplate reader at a wavelength of 450 nm. (Molecular Devices).

### 2.5. Vector Construction and Infection

Lentivirus containing BC200 short hairpin (shBC200) RNA and BC200 overexpression (OEBC200) vectors were purchased from ThermoFisher Scientific (Waltham, MA, USA) and were used according to the manufacturer’s instructions. Two clones of shRNA were used to effectively knockdown (shBC200) and overexpress (OEBC200) BC200; the complete procedure of shRNA lentivirus infection and construction was conducted according to the practice guidelines at a certified BSL-2 laboratory, The Integrated Laboratories for Translational Medicine, Taipei Medical University. Human GB cells were transfected with miR-218-5p (mimic), miR-negative control (miR-NC), and miR-218-5p (inhibitor) were purchased from Qiagen and used as per the manufacturer’s protocol.

### 2.6. Immunohistochemistry

For immunohistochemical (IHC) staining, GB tissue (*n* = 48) and non-tumor brain tissue (*n* = 15) sections (5-μm thick) were obtained from formalin-fixed and paraffin-embedded tissue blocks. Then, the samples were incubated overnight in primary antibodies against mut-p53, Ki-67, MGMT, SOX2, OCT4, BCRP1, MDR1, and MRP1 at 4 °C. The anti-human primary antibodies were shown in Appendix A. After the primary antibodies were washed off, the sections were incubated with goat anti-mouse biotin-conjugated secondary antibodies (1:1000 dilution; Ventana, Oro Valley, AZ, USA) for 20 min at 37 °C. Then, the tissue sections were incubated with streptavidin horseradish peroxidase for 20 min at 37 °C. A 3,3′-diaminobenzidine substrate was applied to the section for 10 min before counterstaining with hematoxylin. The sections in which the primary antibodies were eliminated were used as negative controls.

### 2.7. Western Blot and RT-qPCR

GB cells were washed with PBS and then lysed in RIPA lysis buffer. Cellular protein lysates were isolated using Protein Extraction Kit (Qiagen, Germantown, MD, USA) and quantified using Bradford Protein Assay Kit (Qiagen). In total, 20 μg of samples from different experiments were loaded and subjected to SDS-PAGE using the Mini-Protean III system (Bio-Rad, Taipei City, Taiwan). Separated proteins were transferred onto polyvinylidene fluoride (PVDF) membranes using Trans-Blot Turbo Transfer System (Bio-Rad) followed by blocking with Tris-buffered saline plus skim milk. Then, these PVDF membranes were probed with respective primary antibodies followed by a secondary antibody. The primary antibodies for CD133, KLF4, and SOX2 are shown in Appendix A. ECL detection kit was used for detecting proteins of interest. Images were captured and analyzed using an UVP BioDoc-It system (Analytik Jena, Thuringia, Germany). RT-qPCR was performed using isolated total RNA according to the TRIzol-based protocol (Life Technologies, Carlsbad, CA, USA) provided by the manufacturer. One microgram of total RNA was reverse transcribed using a Qiagen OneStep RT-PCR Kit (Qiagen), and the PCR reaction was performed using a Rotor-Gene SYBR Green PCR Kit (400, Qiagen).

### 2.8. Colony Formation Assay

The colony-forming assay was performed through modification of a previously explained protocol [15]. Briefly, 500 GB cells (BC200, suppressed and overexpressed) were seeded in six-well plates. These cells were allowed to grow for a week and then harvested, fixed, and counted.

### 2.9. Wound Healing Migration Assay

GB cells were seeded in six-well plates (Corning, Corning, NY, USA) with RPMI 1640 medium containing 10% FBS and cultured to 95–100% confluence. Then, a scratch was made along the median axis with a sterile yellow pipette tip across the cells. Cell migration pictures were captured at 0 and 48 h after the medium scratch under a microscope and analyzed with NIH Image J software (https://imagej.nih.gov/ij/download.html).

### 2.10. Matrigel Invasion Assay

GB cells (2 × 10^5^) were seeded in 24-transwell chambers with an 8-μm pore membrane coated with Matrigel in the upper chamber of the transwell system containing serum-free RPMI 1640 medium. The lower chamber of the transwell contained a medium with 20% FBS. After incubation of the cells at 37 °C for 6 h, noninvaded GB cells on the upper side of the membrane were carefully removed with a cotton swab, whereas the invaded cells were stained with crystal violet dye, air-dried, and photographed under a microscope. Images were analyzed using NIH Image J software. (https://imagej.nih.gov/ij/download.html).

### 2.11. Sphere Formation Assay

GB cells (5 × 10^3^ per well) were plated in ultra-low-attachment six-well plates (Corning) containing stem-cell medium consisting of serum-free RPMI 1640 medium supplemented with 10 ng/mL human basic fibroblast growth factor (Invitrogen, Grand Island, NY, USA), 1× B27 supplement (Thermo Fisher Scientific, Carlsbad, CA, USA), and 20 ng/mL epidermal growth factor (Invitrogen). The medium was changed every 72 h. After incubation for 7–14 days, the spheres formed were counted and photographed.

### 2.12. Flow Cytometry

One of the characteristics of cancer stem cells (CSCs) is an increased aldehyde dehydrogenase (ALDH) activity compared with non-CSCs counterparts. In this study, Aldefluor assay kit (StemCell Technologies, Kent, WA, USA) was used to determine the ALDH activity of GB cell lines following the standard protocol. GB cells were removed from the culture dishes with Trypsin-EDTA (Invitrogen), suspended in a buffer containing an ALDH substrate, and incubated at 37 °C for 1.5 h. Flow cytometry was performed using BD LSRFortessa (BD Biosciences, East Rutherford, NJ, USA), and results were analyzed using BD software. Annexin-V was used to detect the drug-induced apoptosis [16]. PE Annexin V and its binding buffer were purchased from Becton-Dickinson. During apoptosis, the phospholipid phosphatidylserine (PS) is translocated from the inner to the outer leaflet of the plasma membrane, thus exposing PS to the external cellular environment. Annexin V has a high affinity for PS, and staining with PE Annexin V serves as a sensitive probe for flow cytometric analysis of cells that are undergoing apoptosis.

### 2.13. In Vivo Studies

An animal study was conducted according to the protocols approved by the Taipei Medical University (Laboratory Animal Center, Affidavit of Approval of Animal Use Protocol, Taipei Medical University, protocol no. LAC-2017-0512). NOD/SCID mice (6–8 weeks old) were purchased from Bio-LASCO Taiwan Co., Ltd. A subcutaneous GB xenograft mouse model was established using tumor spheres grown from GBM8901 cells (1 × 10^6^ cells/20 µL/injection). All treatments started when the tumor became palpable. Mice were then randomly subdivided into four groups: vehicle control (sham injection), TMZ (0.9 mg/kg, p.o., five times/week), shBC200 (where BC200-silenced GBM8901 tumor spheres were injected), and combination of shBC200 with TMZ treatment (0.9 mg/kg, p.o., five times/week). The tumor size was measured using a standard caliper once a week and expressed in cubic centimeters using the formula tumor volume = (length × width^2^)/2, where length represents the longest tumor diameter, and width represents the perpendicular tumor diameter. After the experimental period, mice were sacrificed humanely and tumor samples were collected for further analysis.

### 2.14. Statistical Analysis

All experiments were executed in triplicates. The comparison between two groups was done using the 2-sided Student’s t-test, while one-way analysis of variance (ANOVA) was used to compare ≥3 groups. A *p*-value < 0.05 was considered statistically significant. The Chi-square test were used to evaluate the correlation between BC200 expression and clinicopathological features of primary glioblastoma (GB) patients.

## 3. Results

### 3.1. Upregulation of BC200 RNA Expression in GB Patients

We examined the association of BC200 expression with clinicopathological features of patients with primary GB (*n* = 48). All related data collected are summarized in Table 1. 

Our results showed that BC200 RNA expression have a statistically significant difference between the IDH1 and P53 status. No significant correlation was observed between age and gender. For evaluate the role of BC200 RNA in GB, we initially detected the expression level of BC200 RNA in the blood (whole blood and peripheral blood) of GB and normal patients. Results revealed elevated levels (~2 folds) of BC200 RNA expression in GB patients compared with healthy participants (Figure 1A). Furthermore, the results of IHC analysis of tissue sections obtained from GB patients showed concordance with those of RT-qPCR. Quantitative analysis shows that BC200 RNA expression was higher in 48 GB tissues than in 15 normal adjacent tissues (Figure 1B). The widely used markers to determine the prognostic behavior of brain tumors and survival of GB patients are Ki-67 and p53 [17,18]. Figure 1C,D indicates that high BC200 RNA expression was positively correlated with Ki-67 (*p* < 0.001) and p53-mut expression (*p* < 0.01), suggesting that BC200 RNA is strongly associated with unfavorable prognoses in GB patients.

### 3.2. High BC200 RNA Expression Associated with TMZ Resistance of Stem-Cell-Like Population in GB Cells

We attempted to confirm the role of BC200 RNA expression in chemo-resistance through upregulation of the stem-cell-like population in GB. We first evaluated the differential expression of BC200 RNA in GB (U87MG, DBTRG-05MG, T98G, and GBM8901) and normal human astrocytes (NHA) cells. BC200 RNA expression was significantly higher in GB cell lines than in NHA cells (Figure 2A). The evaluation of cell viability after TMZ treatment revealed that more invasive glioblastoma cells GB cells (GBM8901, T98G) showed higher chemo-resistance (TMZ) than the likely GB cells DBTRG05MG and U87MG cancer cell (Figure 2B). Previous studies have suggested that the increased expression of ALDH+ and CD133+ cells was associated with mesenchymal phenotype and chemoresistance [19], cancer recurrence, and poor prognosis [20]. Furthermore, the role of BC200 RNA in TMZ resistance was assessed through flow cytometric analysis, revealing a significantly higher number of ALDH+/CD133+ cells in cancer cells than in normal human astrocytes (Figure 2C). The expression levels of key markers associated with drug resistance [21,22,23], proliferation, migration, and invasion, such as BCRP1, MDR1, MRP1 and MGMT along with BC200 RNA, were significantly highly expressed in GB cells (Figure 2D).

### 3.3. BC200 RNA Silencing Inhibits the Proliferation, Migration, Invasion, and Self-Renewal Ability of GB Cells

To definitively understand the role of BC200 in GB, we knocked down BC200 RNA and examined its effect on cell migration, motility, and invasion. An analysis using RT-qPCR revealed the transfection efficiency of BC200 shRNA (shBC200 and OEBC200) in GB cells (Figure 3A). All GB cells tested demonstrated a significant reduction in cell migration (Figure 3B), wound healing (Figure 3B), and invasiveness (Figure 3C). Silencing the BC200 RNA significantly inhibited the expression of EMT-associated markers, such as N-cadherin, vimentin, and Slug (Figure 3D). Moreover, the self-renewal capacity was suppressed because of BC200 RNA inhibition, indicated through significant reduction in colony-forming ability (Figure 3E), but no effect on cell proliferation (Figure 3F) and neurosphere generation (Figure 3G) in comparison with mock-transfected control groups. Furthermore, the effect of BC200 RNA inhibition was evaluated on marker expression associated with the self-renewal and pluripotency of GB cells (Figure 3H), such as Oct4, SOX2, and KLF4 [24], which were observed to be downregulated in BC200-suppressed cells in comparison with control mock-transfected cells.

### 3.4. BC200 RNA Overexpression Enhances Aggresivenesss Behevior and Self-Renewal Ability of GB Cells

To further understand the role of BC200 in GB cells, we increased BC200 expression through transfection of a BC200 expression vector OEBC200 in GB cells and used an empty vector as a negative control. Analysis with RT-qPCR showed the transfection efficiency of OEBC200 in GB cells (Figure 4A). BC200 expression was higher in the OEBC200 group than in the control group. Next, BC200 overexpression in GB cell lines demonstrated significantly increased cell wound healing migration (Figure 4B), invasiveness (Figure 4C), and colony-forming ability (Figure 4E) but no effect on GB cell proliferation (Figure 4F) compared with the negative control group. Because of BC200 RNA overexpression, the expression of EMT-associated genes N-cadherin, vimentin, and Slug (Figure 4D) was significantly upregulated. Furthermore, the self-renewal potential of overexpressed BC200 RNA in GB cells compared with mock-transfected cells showed significantly high neurosphere generation (Figure 4G). Moreover, the self-renewal and pluripotency properties of overexpressed BC200 RNA in GB cells (Figure 4H) showed that markers for self-renewal and pluripotency, such as Oct4, SOX2, and KLF4, were upregulated in OEBC200 cells compared with control mock-transfected cells.

### 3.5. BC200 RNA Expression Associated with TMZ Resistance and miR-218-5p Expression

We demonstrated that BC200 RNA expression plays a crucial role in TMZ resistance. To further understand the underlying mechanism of BC200 RNA in GB chemosensitivity, cell viability assay was performed on GB cells with silenced and overexpressed BC200 RNA. The results revealed that the silenced BC200 (shBC200) cells showed decreased TMZ resistance compared with the control group, and OEBC200 cells showed increased TMZ resistance (Figure 5A). Furthermore, we evaluated the expression level of markers associated with drug resistance, proliferation, migration, and invasion in GB cells. Western blot analysis results showed that the levels of O-6-methylguanine-DNA methyltransferase (MGMT), ABCG2 human, ABC transporter (BCRP1), multidrug resistance protein (MDR1), and MRP1 proteins were significantly higher in OEBC200 cells than in BC200-silenced cells (Figure 5B). To further correlate the effect of BC200 RNA on miRNA expression, we applied bioinformatic analysis and observed that miR expression was inversely correlated with BC200 RNA expression. As shown in the heatmap of shBC200 and OEBC200 groups, the level of miR-218-5p was negatively associated with BC200 RNA expression (Figure 5C). The binding prediction of these miRNAs (miR-218-5p) from DIANA and PITA database [25] shows that BC200 RNA is targeted by miR-218-5p (Figure 5D,E). This indicates that a key interaction exists between BC200 RNA and miR-218-5p.

### 3.6. miR-218-5p Modulates Stem Cell Characteristics and TMZ Resistance

Prediction binding of miRNAs (miR-218-5p) from DIANA and PITA database shows it targets BC200 RNA [25]. Thus, we wanted to investigate the relevance of miR-218-5p in controlling the self-renewal potential and TMZ resistance of GB cells. We first verified the expression of miR-218-5p by using RT-qPCR in GB and control cells (Figure 6A). The expression of miR-218-5p was lower in GB cells than in normal human astrocytes. Furthermore, the relative expression of miR-218-5p was observed to be negatively correlated with BC200 RNA expression (Figure 6B) in GB tissues. 

The self-renewal and pluripotency properties of GB cells were analyzed by targeting miR-218-5p. The inhibition of miR-218-5p showed a significantly higher tumorsphere and colony-forming abilities (Figure 6C,D) in GB cells than in the control (NC) mock-transfected groups. Furthermore, the effect of miR-218-5p inhibition was noted on TMZ resistance; miR-218-5p-inhibited cells show higher TMZ resistance than the control mock-transfected groups (Figure 6E) but no significant effect on cell proliferation (Figure 6F). Moreover, markers for self-renewal and ABC Transporters in miR-218-5p-inhibited GB cells (Figure 6G) were evaluated. Markers such as Oct4, SOX2, KLF4, BCRP1, MDR1, and MRP1 were observed to be upregulated in miR-218-5p-inhibited cells compared with control NC and mock-transfected cells. Results revealed that the protein levels of genes associated with Drug resistance, such as MGMT, MLH1, MSH2, MSH6 and PMS2, were significantly higher in the miR-218-5p-inhibited group than in the control NC mock-transfected groups (Figure 6H).

### 3.7. BC200 RNA Inactivity Sensitized GB Cells to TMZ Combined Therapy In Vivo

We evaluated the potential of BC200 RNA knockdown together with TMZ combination treatment in inhibiting the tumor-initiating ability in xenograft models. Tumor was induced in NOD/SCID mice, which were then treated with shBC200, TMZ, control (Vehicle), and combination (shBC200/TMZ) treatments. Tumor volume, body weight, and survival were analyzed for 28 days after cell implantation. At 28 days, mice were sacrificed for immunohistochemistry and gene expression analysis (RT-qPCR). The combination treatment with shBC200 and TMZ led to a significant reduction in tumor size (Figure 7A). Mice treated with shBC200 and the shBC200/TMZ combination showed increased bodyweight but no sign of toxicity and negative treatment effects (Figure 7B). Moreover, the overall survival time was higher in mice treated with the shBC200/TMZ combination compared with that in mice treated with shBC200 and TMZ individually (Figure 7C). Moreover, comparative real-time PCR analyses showed an increased level of miR-218-5p expression in the shBC200/TMZ combination group in comparison with vehicle, TMZ, and shBC200 groups (Figure 7D), suggesting the role of miR-218-5p in GB progression. Furthermore, IHC analysis results of tissue sections in GB patients were consistent with the results of RT-qPCR. IHC results showed that shBC200 and shBC200/TMZ combination groups showed significantly suppressed tumor proliferation, oncogenicity, and tumorigenesis-associated markers (Figure 7E) in tissue samples.

## 4. Discussion

Despite major developments in therapeutic interventions, GB is one of the most lethal and invasive malignancies of the central nervous system and is the leading cause of overall cancer-associated mortality. Patients with GB exhibit poor prognosis [26] and the current therapy mostly fails because these patients develop multidrug resistance and do not respond to the treatment [27,28,29]. Studies have shown that lncRNAs are widely accepted and play a crucial role in the development, progression, and metastasis of cancer as well as in drug resistance of GB. The expression of BC200, a lncRNA, is high in the neuron-specific transcript in the brain [11]. The increased expression of BC200 RNA was atypically found in various human cancers, and its expression was higher in invasive cancers than in benign tumors [13,14]. However, BC200 RNA as a clinical therapeutic target or biomarker for GB is still understudied. In this study, the molecular mechanism of BC200 RNA in glioma development and progression was explored.

In vitro results have suggested that expression of lncRNAs is involved in multiple cellular processes of neurological tumor disease. Regulatory BC200 RNA is highly expressed in the peripheral blood of patients with breast cancer, suggesting that it can be an important molecular tumor biomarker for indicating human malignancies [30]. In our results, BC200 expression have a statistically significant difference in IDH1 and P53 status of GB patients. The overexpression of IDH1 is required to tumor growth in glioblastoma, it could catalyze the conversion of isocitrate and NADP+ to 2-oxoglutarate and NADPH. The most of IDH1/2-mutant brain gliomas affected adults, there is evidence that IDH1/2 mutations are possibly involved with the progression of brain gliomas from grade II to grade III [31]. The different IDH1 mutational status may participate in the development of GBM, IDH1 mutations have been identified an important prognostic marker for GB patients [32,33,34]. In our study, we demonstrated that BC200, both at protein and mRNA level, was significantly upregulated in blood and tissue samples of GB patients (Figure 1). The expression of Ki-67 and p53-mut markers are indicative of prognosis and survival of GB patients [17,18], indicating a positive and significant correlation of BC200 RNA expression with Ki-67and p53. Thus, BC200 RNA is strongly related to poor prognosis of GB patients.

CSCs are now increasingly being recognized as a critical target in cancer treatment because these stem cells are often correlated with the resistance against conventional chemo- and radiotherapy. Overcoming their treatment resistance is the key issue in cancer therapeutics [35,36]. Moreover, studies have reported that dysregulated lncRNA expression remains associated with the TMZ drug resistance of GB. In this study, we observed that the expression of BC200 increased in GB cells compared with normal brain cells (Figure 2). Furthermore, this study showed that GB cells exhibited high expression of ALDH+/CD133+ cells, and key markers associated with drug resistance, proliferation, invasions and metastasis. TMZ resistance is a main reason for treatment fails. The causes of TMZ resistance are mainly DNA repair system, MMR is critical for inducing appropriate cellular responses to DNA damage, previous research suggests that GB cells are TMZ sensitive when MMR is expressed and active [37]. Much of resistance to TMZ observed clinically is due to high expression of MGMT or loss of MMR [38,39]. TMZ treatment induces DNA lesions and the MMR system causing apoptosis in GB [40]. The combination of BC200 inhibition and TMZ treatment may lead to a new therapeutic strategy to improve the efficacy of TMZ in glioblastoma multiforme patients. ABC transporter such as BCRP1, MDR1, and MRP1, were associated with chemoresistance [9]. Moreover, we found that in GB cells, the expression of these markers together with BC200 RNA expression is significantly higher, suggesting their role in TMZ chemoresistance.

To further understand the role of BC200 RNA in GB progression, we used the gain and loss of function of BC200 RNA through gene silencing and overexpression experiments. BC200 knockdown significantly reduced the proliferation, migration, and invasion of GB cells and the expression of markers associated with the self-renewal and pluripotent ability by reducing the colony-forming and neurosphere generation (Figure 3). Meanwhile, BC200 overexpression significantly reduced the BC200 knockdown effect (Figure 4). Therefore, our result indicates that BC200 RNA plays a key role in GB. This result is consistent with those of previous reports indicating the role of BC200 RNA as an oncogene in other cancers, such as breast, cervical, and colon cancers [41,42].

Additionally, to understand the underlying molecular mechanism of BC200 in GB chemosensitivity, a cell viability assay was performed. BC200-silenced cells showed decreased resistance to TMZ compared with the control group (Figure 5A), whereas OEBC200 showed the reverse effect (Figure 5). Western blot analysis showed a reduction in the expression of markers associated with drug resistance in BC200-knockdown cells, and the results were reverse in OEBC200 cells. A previous study indicated that miR in correlation with lncRNA (BC200) expression attenuated the viability, migration, and invasiveness of cancer cells [43]. Subsequently, bioinformatic analysis was performed to evaluate the potential miRNA expression in BC200-inhibited and -overexpressed GB cells. Results from heatmap and binding prediction show that the expression of miR-218-5p is negatively associated with BC200 RNA expression, indicating that miR-218-5p targets BC200 RNA (Figure 5). Studies have shown that an increased expression of miR-218-5p affects cell viability in GB cells [44], but the role of miR-218-5p in conferring chemosensitivity is still not studied. Our result confirms that miR-218-5p is an effective tumor suppressor miRNA that targets BC200 RNA, with high expression in normal cells than in GB cells (Figure 6).

In our model, miR-2018 inhibits MGMT expression, i.e., high BC200 results in low miR-218 and this results in high MGMT and, concomitantly, low toxicity driven by O6-methylguanine adducts that are repaired by MGMT. Additionally, the miR-218-5p plays a key role in preventing the invasiveness of glioma cells. Zhixiao et al. noticed that miR-218-5p can specifically bind to LHFPL3 mRNA and inhibit epithelial-mesenchymal transitions [45]. This is another possible reason to support these results. Furthermore, the relative expression of miR-218-5p was negatively correlated with BC200 RNA expression in GB tissues. In vitro results showed that targeted inhibition of miR-218-5p significantly increased tumorsphere and colony-forming abilities in GB cells compared with the control (NC) mock-transfected groups, confirming TMZ resistance. Again, we observed that markers associated with multidrug resistance, self-renewal, and pluripotency were upregulated in the miR-218-5p-inhibited group, (Figure 6) compared with the control mock-transfected groups, suggesting the role of miR-218-5p in GB cell chemosensitivity, inhibit the expression of the MGMT and MMR system induces resensitization to TMZ. Although the relative expression of ABC-transport protein has regulated by BC200/mir-218-5p axis, but ABC-transport protein as inducers of the TMZ resistant phenotype in GB is still controversial [46]. Finally, we demonstrated that the combination of BC200 inhibition and TMZ treatment increased GB sensitivity. It greatly inhibited the tumor growth in the xenograft mouse model with no negative impact on body weight and survival (Figure 7). The results of IHC analysis are concordant with those of RT-qPCR; the shBC200/TMZ combination group shows significant suppression of tumor proliferation, oncogenicity, and tumorigenesis-associated markers. Moreover, the miR-218-5p level expressed was significantly high in the combination treatment group. Collectively, we provided a strong preclinical evidence in support of using shBC200/TMZ combination for treating malignant glioma cancer. In addition, miR-218-5p could be used as a biomarker for monitoring therapeutic responses.

## 5. Conclusions

TMZ resistance is one of the critical causes of treatment failure in GB patients. The molecular mechanisms of TMZ resistance are still unclear. Hence, we used GB cell lines exposed to TMZ treatment to analyze the relevance of MGMT and MMR system between BC200/miR-218-5p axis in the TMZ resistance phenomenon. In summary, as shown in schema abstract of Figure 8, our results suggested that BC200 RNA, a lncRNA, is highly expressed both in vitro and in vivo and significantly modulates GB oncogenicity and enhances TMZ chemoresistance through concomitantly enhancing self-renewal and pluripotency of GB cells by modulating the expression of tumor-suppressor miR-218-5p. MiR-218-5p effectively targets and inhibits self-renewal and ABC Transporters markers such as Oct4, SOX2, BCRP1, MDR1, MRP1, and MGMT. Resultant in the reduction of TMZ-resistance. Our findings highlight the therapeutic efficacy of BC200 RNA as a clinical biomarker or therapeutic target for GB.

## Figures and Tables

**Figure 1 cells-09-01859-f001:**
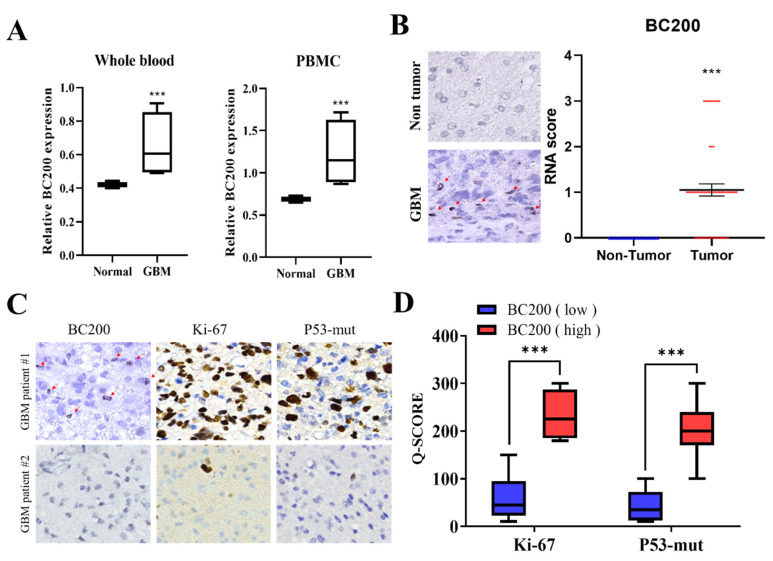
Expression level of full-length BC200 RNA in the blood and tissue. (**A**) Whole blood or PBMC from healthy participants and GB patients were assessed using RT-qPCR. GAPDH was used as the endogenous control. (**B**) The differential expression of BC200 in GB samples (*n* = 48) and adjacent normal tissues (*n* = 15) is shown. (**C**, **D**) The differential expression of Ki-67 and p53-mut in GB patients (*n* = 48) with a high or low expression of BC200 RNA. *** *p* < 0.001.

**Figure 2 cells-09-01859-f002:**
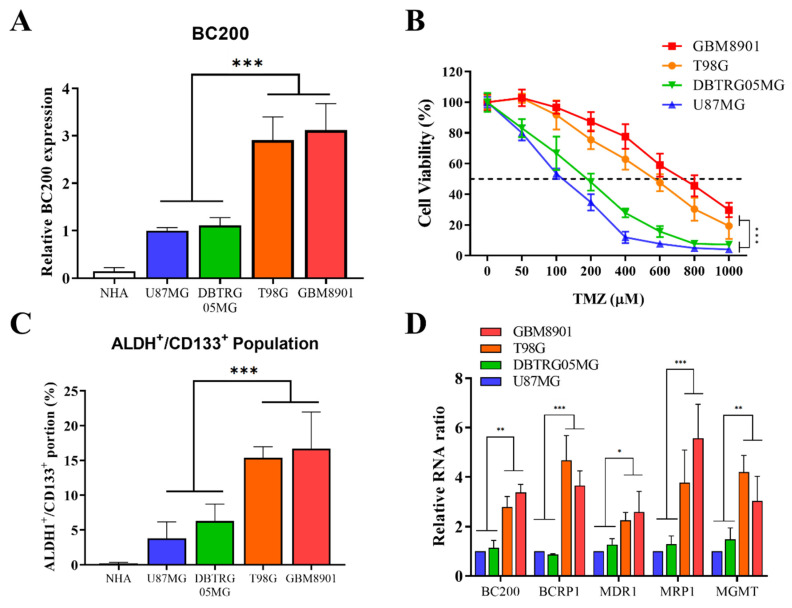
Differential expression of BC200 RNA in GB cell lines. (**A**) The differential expression of BC200 RNA in GB cell lines and normal human astrocytes are shown. (**B**) The viability of GB cells was analyzed through SRB assay 48 h after TMZ (0–1000 μM) treatment. (**C**) Flow cytometry analysis of the ALDH1^+^/CD133^+^ portion in GB cell lines and normal human astrocytes. (**D**) The level of BC200, BCRP1, MDR1, MRP1 and MGMT in GB cell lines was analyzed using RT-qPCR. * *p* < 0.05, ** *p* < 0.01, and *** *p* < 0.001.

**Figure 3 cells-09-01859-f003:**
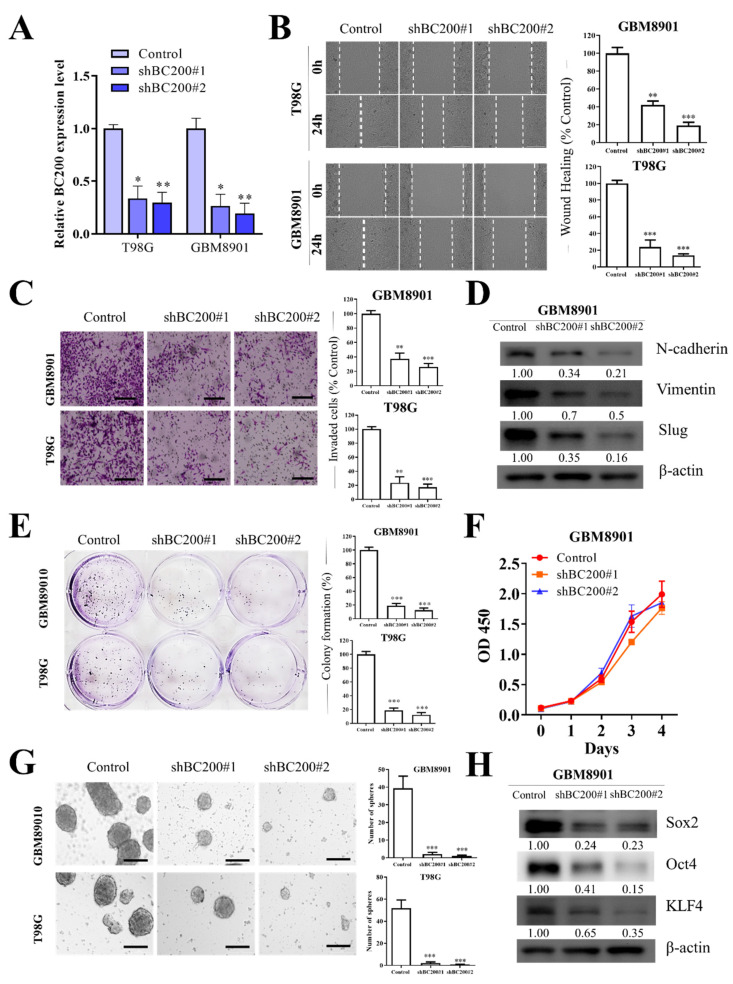
BC200 knockdown inhibited cell migration, invasion, colony formation, and sphere formation in GB cells in vitro. (**A**) The level of BC200 transfection with shBC200 was analyzed using RT-qPCR. (**B**) Wound-healing assay or migration assay showed that shBC200 resulted in delayed healing of the scratch wound. (**C**) Transwell invasion assay was used and results were expressed as the number of invaded cells per field. (**D**) The levels of N-cadherin, vimentin, and slug following shBC200 in GB cells were determined through western blot. (**E**) Colony formation assays showed that BC200 knockdown inhibited GB cell survival. (**F**) CCK-8 assay showed that BC200 knockdown had no effect on GB cell proliferation. (**G**) Neurosphere formation assays showed that BC200 knockdown inhibited GB cell stemness. (**H**) The levels of SOX2, Oct4, and KLF4 following shBC200 transfection in GB cells were determined through western blot analysis. * *p* < 0.05, ** *p* < 0.01, and *** *p* < 0.001.

**Figure 4 cells-09-01859-f004:**
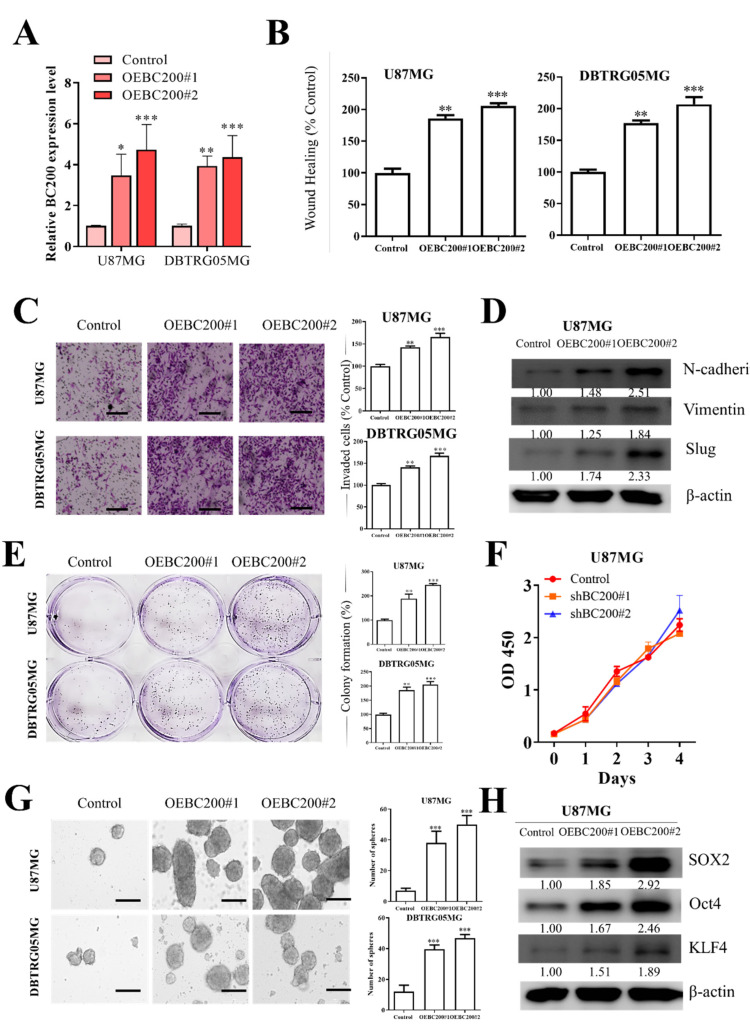
BC200 RNA overexpression promotes cell migration, invasion, colony formation, and sphere formation on GB cells in vitro. (**A**) The level of BC200 transfection with OEBC200, an expression vector, was analyzed using RT-qPCR. (**B**) Migration assay showed that OEBC200 resulted in fast scratch wound healing. (**C**) Transwell invasion assay was performed, and the results were expressed as the number of invaded cells per field. (**D**) Following OEBC200 treatment, the levels of N-cadherin, vimentin, and slug in GB cells were determined through western blot. (**E**) Colony formation assays showed that BC200 RNA overexpression enhances GB cell survival. (**F**) CCK-8 assay showed that OEBC200 had no effect on GB cell proliferation. (**G**) Neurosphere formation assays showed that a high expression of BC200 promotes GB cell stemness. (**H**) Following OEBC200 treatment, the levels of SOX2, Oct4, and KLF4 in GB cells were determined through western blot. * *p* < 0.05, ** *p* < 0.01, and *** *p* < 0.001.

**Figure 5 cells-09-01859-f005:**
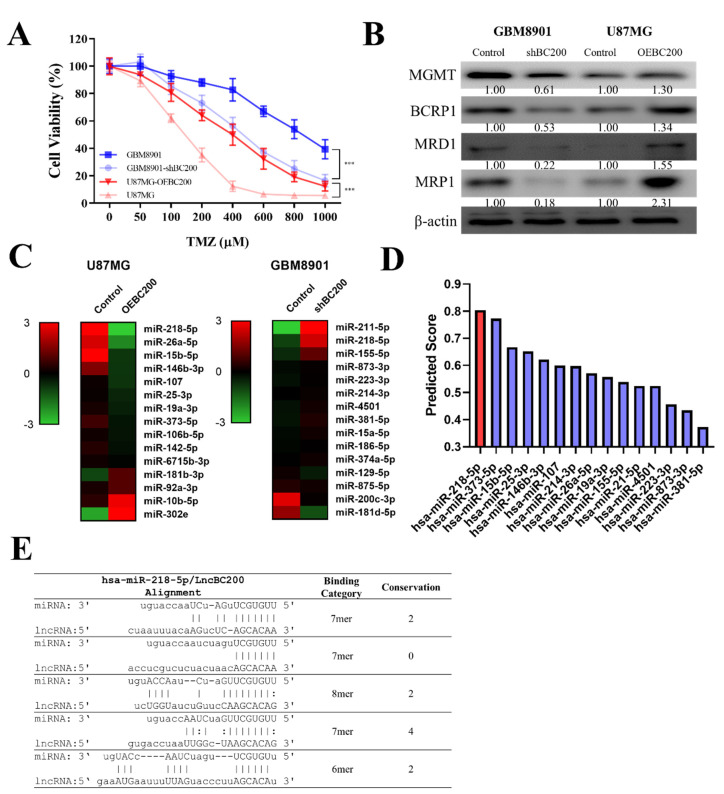
BC200 RNA promotes TMZ resistance in GB through sponge miR-218-5p. (**A**) The viability of shBC200 and OEBC200 GB cell lines was analyzed through SRB assay 48 h after TMZ (0–1000 μM) treatment. (**B**) The levels of MGMT, BCRP1, MDR1, and MRP1 following shBC200 and OEBC200 in GB cells were determined through western blot. (**C**) MicroRNA profiling analyses showed that shBC200 and OEBC200 contained high and low levels of miR-218-5p, respectively. (**D**) LncBase Predicted v.2 predicted that a high binding score of miR-218-5p with BC200. (**E**) BC200 directly interacts with multiple binding sites to hsa-miR-218-5p. *** *p* < 0.001.

**Figure 6 cells-09-01859-f006:**
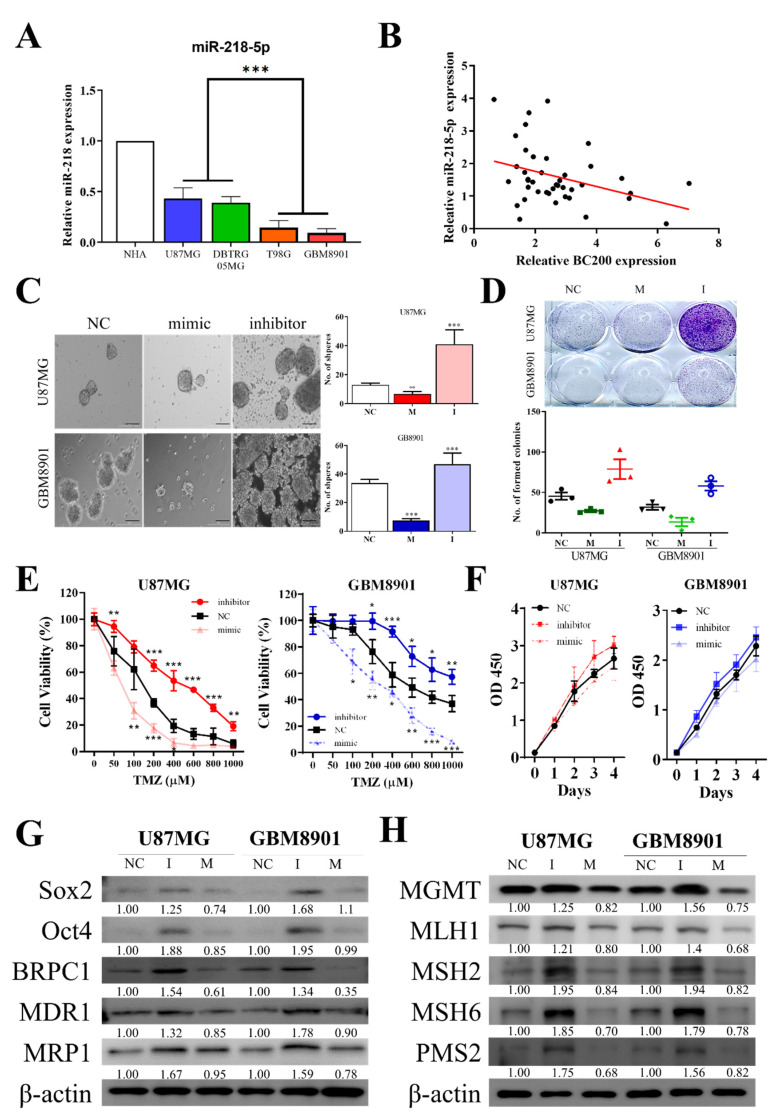
miR-218-5p regulated cell sphere formation, colony formation, and TMZ resistance in GB cells in vitro. (**A**) The differential expression of miR-218-5p in GB cell lines and normal human astrocytes. (**B**) Pearson’s correlation curve identified a negative correlation between BC200 and miR-218-5p in GB tissues. (**C**) Sphere formation assays showed that inhibition or mock transfection of miR-218-5p regulated GB cell stemness. (**D**) Colony formation assays showed that inhibition or mock transfection of miR-218-5p regulated GB cell survival. (**E**) The viability of GB cells with inhibition or mock transfection of miR-218-5p was analyzed through SRB assay 48 h after TMZ (0–1000 μM) treatment. (**F**) CCK-8 assay showed that inhibition or mock transfection of miR-218-5p had no effect on GB cell proliferation. (**G**) The protein levels of SOX2, Oct4, BRPC1, MRP1 and MDR1 with inhibition or mock transfection of miR-218-5p. (**H**) The protein levels of MGMT, MLH1, MSH2, MSH6 and PMS2 following inhibition or mock transfection of miR-218-5p in GB cells were determined through western blot. * *p* < 0.05, ** *p* < 0.01 and *** *p* < 0.001.

**Figure 7 cells-09-01859-f007:**
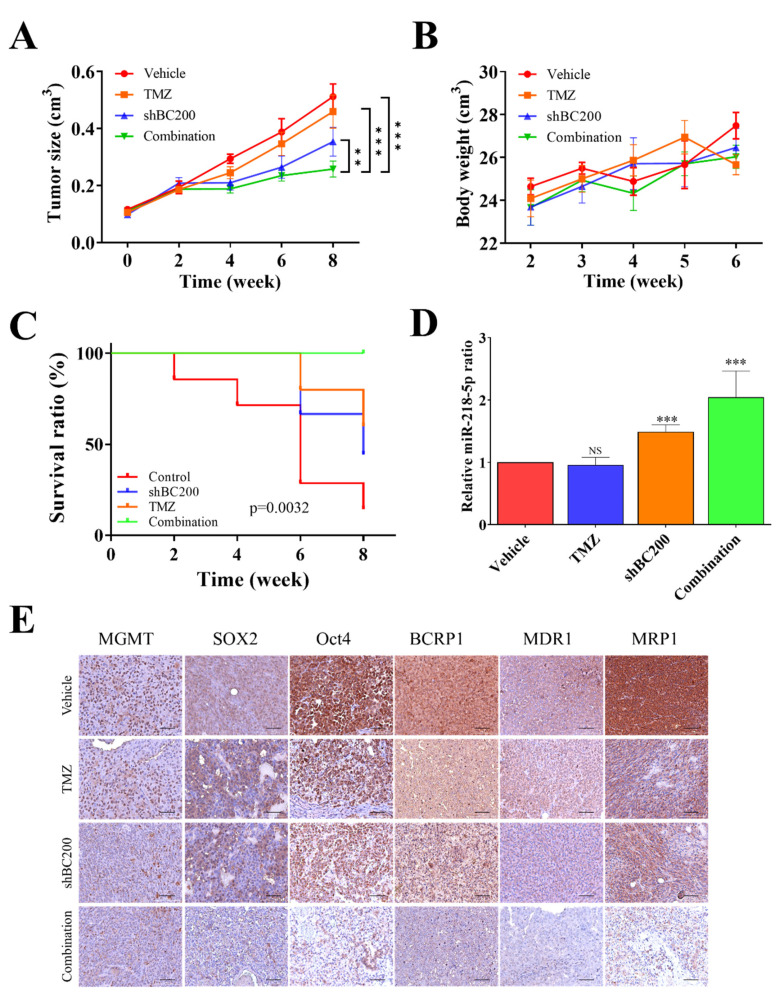
Combination of shBC200 and TMZ inhibits tumor growth in GB mice models. (**A**) Tumor size over time curve indicates that the most significantly delayed tumor growth was observed in the shBC200 and TMZ combination group, followed by shBC200 group, whereas TMZ and control groups did not differ significantly. (**B**) Bodyweight curves over time suggest no clear cytotoxic effects in all mice because no significant decrease in weight was observed. (**C**) Kaplan–Meier survival curve showed increased median overall survival in TMZ, shBC200, and shBC200/TMZ combination, whereas the combination group showed the best survival ratio. (**D**) Comparative real-time PCR analyses showed a significantly increased mRNA level of miR-218-5p expression in both shBC200 and shBC200/TMZ combination groups as compared with vehicle and TMZ counterparts. (**E**) IHC staining showed that treatment in the shBC200 group and shBC200/TMZ combination group suppressed tumorigenesis and TMZ resistance. ** *p* < 0.01 and *** *p* < 0.001.

**Figure 8 cells-09-01859-f008:**
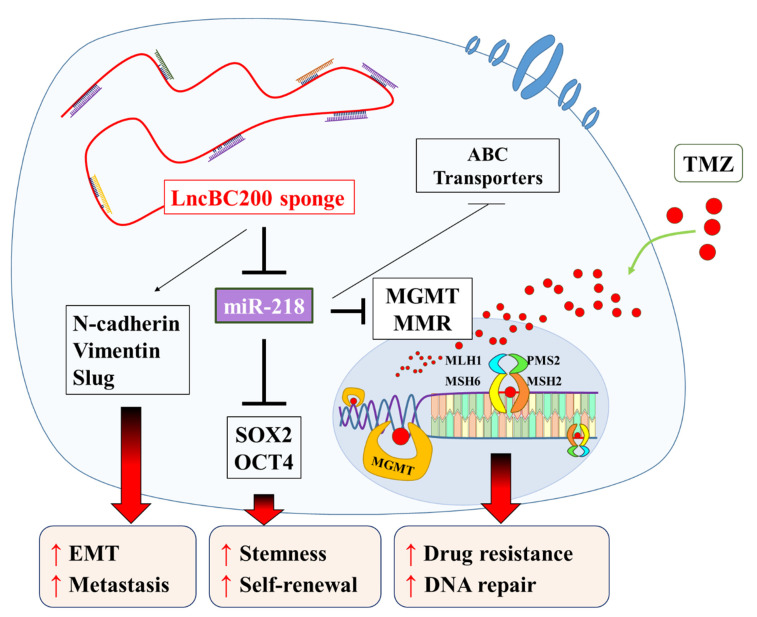
The lncRNA BC200 RNA sponges miR-218-5p regulated MGMT and MMR system enhancing self-renewal and TMZ resistance of GBM cells.

**Table 1 cells-09-01859-t001:** Analysis of the association of BC200 expression with clinicopathological features of patients with primary GB (*n* = 48).

	High BC200 n=25 (52%)	Low BC200 n=23 (48%)	*p*-Value
Age			
<65	15 (60%)	8 (35%)	0.0806
≥65	10 (40%)	15 (65%)	
Gender			
male	14 (56%)	13 (57%)	0.971
female	11 (44%)	10 (43%)	
IDH1 status			
wildtype	24 (96%)	17 (74%)	0.0303
mutation	1 (4%)	6 (26%)	
P53 status			
wildtype	9 (36%)	15 (65%)	0.0431
mutation	16 (64%)	8 (35%)	

## Data Availability

The datasets used and analyzed in the current study are publicly-accessible as indicated in the manuscript.

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
