# Peer review of "Targeting BC200/miR218-5p Signaling Axis for Overcoming Temozolomide Resistance and Suppressing Glioma Stemness"

_cells, 2020, doi:10.3390/cells9081859_

Round 1

Reviewer 1 Report

In their manuscript “Targeting BC200/miR218-5p signaling axis for overcoming temozolomide resistance and suppressing glioma stemness” Yu-Kai Su and colleagues lay out a very nice case for a putative role of BC200 and its antagonist miR-218-5p in Glioblastoma biology. On the whole the paper is interesting and well-written, but some issues should be further clarified.

Major points:

  1. Figure 2: The concentrations of TMZ used are far beyond the physiological relevant concentrations and it is highly debateable whether high concentrations mimic repeated exposure to low concentrations. Also, cell viability (not defined in Materials & Methods, assumingly something akin to MTT?) is also known to give weird read outs, stem cell-like cells are often more sensitive than differentiated cells, which is not reflected in cell death and only partially in changes in cell numbers (both therapeutically more relevant read outs, unless the authors dispute that TMZ is primarily given as an apoptosis-inducing agent, which is a valid dispute, but makes things even more complicated). Furthermore, ALDH1 positive cells are not necessarily stem cells as implied in the Abstract. Finally, cell lines should be checked for both Ki-67 and p53 (p53 is wt in U87 and mutant in T98, so protein is actually higher in T98, but not functional. It is by the by unclear whether the authors looked at p53 expression (as implied in the text) or mutational status (implied in the figure) in figure 1.
  2. Figure 3 and 4: It should be shown if proliferation rate population doubling time is altered, differences would explain whole data set (not making results less important, just showing a single role for BC200 with multiple effects).
  3. Figure 5B: Wouldn’t components of the DNA repair machinery be more important to look at in terms of TMZ resistance, plus to show that MGMT expression is (most likely) not affected?
  4. Unclear, as of Figure 6, why transfecting shBC200 or BC200mimetic should affect miR-218-5p, if miR-218-5p is the regulator of BC200. I assume the author imply a negative feedback loop, but show no evidence for this. Figure 8 (which is a very good addition to the paper and a nice summary) is rather speculative regarding the relationships of the individual players…
  5. While classical EMT does not occur in GB, as no basement membrane is present and E-cadherin is only rarely expressed in GB (Iwadate, 2016), there is considerable overlap between the gene expression profile of EMT (from human mammary epithelial cells) and (not just the mesenchymal subtype of) GB (Zarkoob et al., 2013), i.e. GB is always invasive!
  6. Also Figure 6, proliferation should again be considered.
  7. Figure 7: Tumour wasn’t “induced”, that term is better reserved for chemical induction and the use of survival curves for subcutaneous models is rather debateable. Also terminating the survival curve at 8 weeks is far too early to make any significant statement regarding survival.
  8. The Student’s t-test is inappropriate for analysing most data sets presented. While this will not alter the overall message of the paper, it should be addressed.

Minor points:

  1. You mainly call the tumour (correctly) Glioblastoma, so use GB as abbreviation, see below
  2. Delete “multiforme” from abstract, superfluous.
  3. Abstract, line 50: “noncoding RNA BC200/miR-218-5p signaling circuit” format changed
  4. Background: GB is a grade IV astrocytoma, there are no Glioblastoma Grade I-III
  5. Also reference 2 is out of date, replace with Acta Neuropathologica volume 131, pages803–820(2016)
  6. Background, lines 65/66: Recent used too often, inelegant
  7. Background, lines 67f: This is phrased as if lncRNAs contribute to GB metastasis, which is an event that is of almost negligible importance in GB, as it occurs in less than two percent
  8. Background, lines 74f: BC200 is expressed “in the presence of neurodegenerative diseases”, isn’t that rather surprising? These diseases are not characterised by motile cells and if anything, are associated with increased cell death, not resistance to cell death.
  9. Why was RPMI medium used for the cell lines, ATCC recommends DMEM
  10. Results, line 214: GB patients and normal controls
  11. Results, line 238: “the likely GBM cells” unclear
  12. Results, line 245: “significantly highly” is ungrammatical and unclear, expressed higher in two of four GB cells.
  13. Figure 3: If you define wound closure of controls as 100% you should not have any StDev on the control bar, ditto for invasion and colony forming assays.
  14. Results, line 293: significantly high(er)
  15. Discussion, line 402: “Studies have shown that lncRNAs are widely accepted” unclear, are author arguing that the existence of lncRNA is no longer debated? Wasn’t aware that was ever the case.
  16. Discussion, line 420ff: Both, the existence of stem cells in GB and their role in TMZ-resistance are not as straight forward as with other tumours and should be discussed a bit more…
  17. Misregulated or dysregulated?
  18. Discussion, line 430f: significant high is not a good term, see other comments regarding this.
  19. Discussion, line 443: (Figure 5A) should be bold
  20. Lines 494ff: Font is wrong for some names

Author Response

Point-by-point responses to reviewer's comments:

We would like to thank all the reviewers for the thorough reading of our manuscript as well as their valuable comments. We have followed the reviewer's comments thoroughly and feel that they have further helped in strengthening the manuscript.

Reviewer 1 Comments and Suggestions for Authors

In their manuscript “Targeting BC200/miR218-5p signaling axis for overcoming temozolomide resistance and suppressing glioma stemness” Yu-Kai Su and colleagues lay out a very nice case for a putative role of BC200 and its antagonist miR-218-5p in Glioblastoma biology. On the whole the paper is interesting and well-written, but some issues should be further clarified.

  1. Figure 2: The concentrations of TMZ used are far beyond the physiological relevant concentrations and it is highly debateable whether high concentrations mimic repeated exposure to low concentrations. Also, cell viability (not defined in Materials & Methods, assumingly something akin to MTT?) is also known to give weird read outs, stem cell-like cells are often more sensitive than differentiated cells, which is not reflected in cell death and only partially in changes in cell numbers (both therapeutically more relevant read outs, unless the authors dispute that TMZ is primarily given as an apoptosis-inducing agent, which is a valid dispute, but makes things even more complicated). Furthermore, ALDH1 positive cells are not necessarily stem cells as implied in the Abstract.

R1: The reviewer's comments are much appreciated. As per the concentration of TMZ, we followed the conditions described by Borhani S, (2017). In Figure 2B, we are looking at and evaluating the effect of TMZ treatment on GBM cells. GBM cells (GBM8901 and T98G) had shown higher chemo-resistance (TMZ) than the GBM likely cells (DBTRG05MG and U87MG cells). Moreover, we have changed Figure 2 accordingly with additional FACS analysis using both ALDH1 and CD133 for the verification of cancer stem-like properties in updated Figure 2. Please see the updated results section, line 253-269.

3.2 High BC200 RNA expression associated with TMZ resistance of stem-cell-like population in GBM cells

We attempted to confirm the role of BC200 RNA expression in chemo-resistance through upregulation of the stem-cell-like population in GBM. We first evaluated the differential expression of BC200 RNA in GBM (U87MG, DBTRG-05MG, T98G, and GBM8901) and normal human astrocytes (NHA) cells. BC200 RNA expression was significantly higher in GBM cell lines than in NHA cells (Figure 2A). The evaluation of cell viability after TMZ treatment revealed that GBM cells showed higher chemo-resistance (TMZ) than the likely GBM cells (Figure 2B). Previous studies have suggested that the increased expression of ALDH+ cells was associated with chemoresistance [19], cancer recurrence, and poor prognosis [20]. Furthermore, the role of BC200 RNA in TMZ chemoresistance was assessed through flow cytometric analysis, revealing a significantly higher number of ALDH+/CD133+ cells in cancer cells than in normal human astrocytes (Figure 2C). The expression levels of key markers associated with drug resistance, proliferation, migration, and invasion, such as BCRP1 [21], MDR1 [22], and MRP1 [23] along with BC200 RNA, were significantly highly expressed in GBM cells (Figure 2D).

Please also see the updated figure 2 legend, line 271-275.

Figure 2. Differential expression of BC200 RNA in GBM cell lines. (A) The differential expression of BC200 RNA in GBM cell lines and normal human astrocytes are shown. (B) The viability of GBM cells was analyzed through SRB assay 48 h after TMZ (0–1000 μM) treatment. (C) Flow cytometry analysis of the ALDH1+/CD133+ portion in GBM cell lines and normal human astrocytes. (D) The level of BC200, BCRP1, MDR1, and MRP1 in GBM cell lines was analyzed using qRT-PCR. *P < 0.05, **P < 0.01, and ***P < 0.001.

We have included the cell viability assay protocol in the revised material and method section. Please see line 111-128.

2.3 Sulforhodamine B (SRB) Viability Assay

T98G, U87MG, DBTRG-05MG and GB8401 cells were seeded in 96-well plates in triplicates at a concentration of 3,000 cells per well. After 24 h incubation in a 5% CO2 humidified incubator at 37 °C, the cells were treated with varying concentrations of 0–1000 μM TMZ as indicated for 24 h. Thereafter, cells were washed in PBS twice, fixed in cold 10% trichloroacetic acid (TCA) for 1h, washed with distilled water, and then incubated in 0.4 SRB (w/v) in 1% acetic acid at room temperature for 1 h. After washing of unbound SRB dye with 1% acetic acid thrice, the plates were air-dried. Attached dye was dissolved in 20 mM trizma base, and absorbance was read in a microplate reader at a wavelength of 570 nm. (Molecular Devices, Sunnyvale, CA, USA).

2.4 Cell proliferation Assay

Cell Counting Kit-8 (CCK-8, Dojindo Laboratories, Rockville, MD, USA) was applied for detecting the cell proliferation. T98G, U87MG, DBTRG-05MG and GB8401 cells were seeded in 96-well plates in triplicates, incubated for 24, 48, 72 or 96h. For each time point, 10μL CCK-8 solution was added per well and the cells were incubated at 37°C for 2h and absorbance was read in a microplate reader at a wavelength of 450 nm. (Molecular Devices, Sunnyvale, CA, USA).

  1. Finally, cell lines should be checked for both Ki-67 and p53 (p53 is wt in U87 and mutant in T98, so protein is actually higher in T98, but not functional. It is by the by unclear whether the authors looked at p53 expression (as implied in the text) or mutational status (implied in the figure) in figure 1.

R2: We thank the reviewer for the comments and apologize for the confusion. In Figure 1, we are evaluating the expression level of BC200 in GBM and normal patients. For that we initially detected the expression level of BC200 RNA in the blood (Whole blood and Peripheral blood; PBMC) of GBM patients and normal controls. In Fig 1C-D the correlation analysis of the expression of BC200 with Ki-67 and p53-mut has been shown. The BC200, higher expressing GBM patient's samples showed higher expression of ki-67 and p53-mut in comparison to the BC200 low expressing patients. Please see updated Figure 1, line 235-246.

3.1 Upregulation of BC200 RNA expression in GBM patients

To evaluate the role of BC200 RNA in GBM, we initially detected the expression level of BC200 RNA in the blood (whole blood and peripheral blood) of GBM and normal patients. Results revealed elevated levels (~2 folds) of BC200 RNA expression in GBM patients compared with healthy participants (Figure 1A). Furthermore, the results of IHC analysis of tissue sections obtained from GBM patients showed concordance with those of qRT-PCR. Quantitative analysis shows that BC200 RNA expression was higher in 48 GBM tissues than in 15 normal adjacent tissues (Figure 1B). The widely used markers to determine the prognostic behavior of brain tumors and survival of GBM patients are Ki-67 and p53 [17, 18]. Figure 1C and 1D indicates that high BC200 RNA expression was positively correlated with Ki-67 (p<0.001) and p53-mut expression (p<0.01), suggesting that BC200 RNA is strongly associated with unfavorable prognoses in GBM patients.

Please also see the updated figure 1 legend, line 248-252.

Figure 1. Expression level of full-length BC200 RNA in the blood and tissue. (A) Whole blood or PBMC from healthy participants and GBM patients were assessed using qRT-PCR. GAPDH was used as the endogenous control. (B) The differential expression of BC200 in GBM samples (n = 48) and adjacent normal tissues (n = 15) is shown. (C and D) The differential expression of Ki-67 and p53-mut in GBM patients with a high or low expression of BC200 RNA. *P < 0.05, **P < 0.01, and ***P < 0.001.

  1. Figure 3 and 4: It should be shown if proliferation rate population doubling time is altered, differences would explain whole data set (not making results less important, just showing a single role for BC200 with multiple effects).

R3: The reviewer's comments are greatly appreciated. We have included the proliferation and doubling time of shBC200/OEBC200 transfected GBM cells in updated Figure 3F and Figure 4F. Please also kindly see updated results section, line 276-291 and line 303-319.

3.3 BC200 RNA silencing inhibits the proliferation, migration, invasion, and self-renewal ability of GBM cells

To definitively understand the role of BC200 in GBM, we knocked down BC200 RNA and examined its effect on cell migration, motility, and invasion. An analysis using qRT-PCR revealed the transfection efficiency of BC200 shRNA (shBC200 and OEBC200) in GBM cells (Figure 3A). All GBM cells tested demonstrated a significant reduction in cell migration (Figure 3B), wound healing (Figure 3B), and invasiveness (Figure 3C). Silencing the BC200 RNA significantly inhibited the expression of EMT-associated markers, such as N-cadherin, vimentin, and Slug (Figure 3D). Moreover, the self-renewal capacity was suppressed because of BC200 RNA inhibition, indicated through significant reduction in colony-forming ability (Figure 3E), but no effect on cell proliferation (Figure 3F) and neurosphere generation (Figure 3G) in comparison with mock-transfected control groups.  Furthermore, the effect of BC200 RNA inhibition was evaluated on marker expression associated with the self-renewal and pluripotency of GBM cells (Figure 3H), such as Oct4, SOX2, and KLF4 [24], which were observed to be downregulated in BC200-suppressed cells in comparison with control mock-transfected cells.

3.4 BC200 RNA overexpression enhances aggresivenesss behevior and self-renewal ability of GBM cells

To further understand the role of BC200 in GBM cells, we increased BC200 expression through transfection of a BC200 expression vector OEBC200 in GBM cells and used an empty vector as a negative control. Analysis with qRT-PCR showed the transfection efficiency of OEBC200 in GBM cells (Figure 4A). BC200 expression was higher in the OEBC200 group than in the control group. Next, BC200 overexpression in GBM cell lines demonstrated significantly increased cell wound healing migration (Figure 4B), invasiveness (Figure 4C), and colony-forming ability (Figure 4E) but no effect on GBM cell proliferation (Figure 4F) compared with the negative control group. Because of BC200 RNA overexpression, the expression of EMT-associated genes N-cadherin, vimentin, and Slug (Figure 4D) was significantly upregulated. Furthermore, the self-renewal potential of overexpressed BC200 RNA in GBM cells compared with mock-transfected cells showed significantly high neurosphere generation (Figure 4G). Moreover, the self-renewal and pluripotency properties of overexpressed BC200 RNA in GBM cells (Figure 4H) showed that markers for self-renewal and pluripotency, such as Oct4, SOX2, and KLF4, were upregulated in OEBC200 cells compared with control mock-transfected cells.

Please also see the updated figure 3 legend, line 293-301.

Figure 3. BC200 knockdown inhibited cell migration, invasion, colony formation, and sphere formation in GB cells in vitro. (A) The level of BC200 transfection with shBC200 was analyzed using qRT-PCR. (B) Wound-healing assay or migration assay showed that shBC200 resulted in delayed healing of the scratch wound. (C) Transwell invasion assay was used and results were expressed as the number of invaded cells per field. (D) The levels of N-cadherin, vimentin, and slug following shBC200 in GB cells were determined through western blot. (E) Colony formation assays showed that BC200 knockdown inhibited GB cell survival. (F) CCK-8 assay showed that BC200 knockdown had no effect on GB cell proliferation. (G) Neurosphere formation assays showed that BC200 knockdown inhibited GB cell stemness. (H) The levels of SOX2, Oct4, and KLF4 following shBC200 transfection in GB cells were determined through Western blot analysis. *P < 0.05, **P < 0.01, and ***P < 0.001.

Please also see the updated figure 4 legend, line 321-330.

Figure 4. BC200 RNA overexpression promotes cell migration, invasion, colony formation, and sphere formation on GB cells in vitro. (A) The level of BC200 transfection with OEBC200, an expression vector, was analyzed using qRT-PCR. (B) Migration assay showed that OEBC200 resulted in fast scratch wound healing. (C) Transwell invasion assay was performed, and the results were expressed as the number of invaded cells per field. (D) Following OEBC200 treatment, the levels of N-cadherin, vimentin, and slug in GB cells were determined through western blot. (E) Colony formation assays showed that BC200 RNA overexpression enhances GB cell survival. (F) CCK-8 assay showed that OEBC200 had no effect on GB cell proliferation. (G) Neurosphere formation assays showed that a high expression of BC200 promotes GB cell stemness. (H) Following OEBC200 treatment, the levels of SOX2, Oct4, and KLF4 in GB cells were determined through western blot. *P < 0.05, **P < 0.01, and ***P < 0.001.

  1. Figure 5B: Wouldn't components of the DNA repair machinery be more important to look at in terms of TMZ resistance, plus to show that MGMT expression is (most likely) not affected?

R4: Reviewer's comments are greatly appreciated. We have included the expression of MGMT in updated Figure 2D and Figure 5B. Please kindly see revised results section, line 253-269 and line 331-350.

3.2 High BC200 RNA expression associated with TMZ resistance of stem-cell-like population in GBM cells

We attempted to confirm the role of BC200 RNA expression in chemo-resistance through upregulation of the stem-cell-like population in GBM. We first evaluated the differential expression of BC200 RNA in GBM (U87MG, DBTRG-05MG, T98G, and GBM8901) and normal human astrocytes (NHA) cells. BC200 RNA expression was significantly higher in GBM cell lines than in NHA cells (Figure 2A). The evaluation of cell viability after TMZ treatment revealed that GBM cells showed higher chemo-resistance (TMZ) than the likely GBM cells (Figure 2B). Previous studies have suggested that the increased expression of ALDH+ and CD133+ cells was associated with mesenchymal phenotype and chemoresistance [19], cancer recurrence, and poor prognosis [20]. Furthermore, the role of BC200 RNA in TMZ chemoresistance was assessed through flow cytometric analysis, revealing a significantly higher number of ALDH+/CD133+ cells in cancer cells than in normal human astrocytes (Figure 2C). The expression levels of key markers associated with drug resistance [21, 22, 23], proliferation, migration, and invasion, such as BCRP1, MDR1, MRP1 and MGMT along with BC200 RNA, were significantly highly expressed in GBM cells (Figure 2D).

3.5 BC200 RNA expression associated with TMZ chemoresistance and miR-218-5p expression

We demonstrated that BC200 RNA expression plays a crucial role in TMZ resistance. To further understand the underlying mechanism of BC200 RNA in GBM chemosensitivity, cell viability assay was performed on GBM cells with silenced and overexpressed BC200 RNA. The results revealed that the silenced BC200 (shBC200) cells showed decreased TMZ resistance compared with the control group, and OEBC200 cells showed increased TMZ chemoresistance (Figure 5A). Furthermore, we evaluated the expression level of markers associated with drug resistance, proliferation, migration, and invasion in GBM cells. Western blot analysis results showed that the levels of MGMT (O-6-Methylguanine-DNA Methyltransferase), BCRP1 (ABCG2 human, ABC transporter), MDR1 (Multidrug Resistance Protein), and MRP1 proteins were significantly higher in OEBC200 cells than in BC200-silenced cells (Figure 5B). To further correlate the effect of BC200 RNA on miRNA expression, we applied bioinformatic analysis and observed that miR expression was inversely correlated with BC200 RNA expression. As shown in the heatmap of shBC200 and OEBC200 groups, the level of miR-218-5p was negatively associated with BC200 RNA expression (Figure 5C). The binding prediction of these miRNAs (miR-218-5p) from DIANA and PITA database [25] shows that BC200 RNA is targeted by miR-218-5p (Figure 5D and E). This indicates that a key interaction exists between BC200 RNA and miR-218-5p.

Please also see the updated figure 2 legend, line 271-275.

Figure 2. Differential expression of BC200 RNA in GBM cell lines. (A) The differential expression of BC200 RNA in GBM cell lines and normal human astrocytes are shown. (B) The viability of GBM cells was analyzed through SRB assay 48 h after TMZ (0–1000 μM) treatment. (C) Flow cytometry analysis of the ALDH1+/CD133+ portion in GBM cell lines and normal human astrocytes. (D) The level of BC200, BCRP1, MDR1, MRP1 and MGMT in GBM cell lines was analyzed using qRT-PCR. *P < 0.05, **P < 0.01, and ***P < 0.001.

Please also see the updated figure 5 legend, line 352-358.

Figure 5. BC200 RNA promotes TMZ resistance in GBM through sponge miR-218-5p. (A) The viability of shBC200 and OEBC200 GBM cell lines was analyzed through SRB assay 48 h after TMZ (0–1000 μM) treatment. (B) The levels of MGMT, BCRP1, MDR1, and MRP1 following shBC200 and OEBC200 in GBM cells were determined through western blot. (C) MicroRNA profiling analyses showed that shBC200 and OEBC200 contained high and low levels of miR-218-5p, respectively. (D) LncBase Predicted v.2 predicted that a high binding score of miR-218-5p with BC200. (E) BC200 directly interacts with multiple binding sites to hsa-miR-218-5p.

  1. Unclear, as of Figure 6, why transfecting shBC200 or BC200mimetic should affect miR-218-5p, if miR-218-5p is the regulator of BC200. I assume the author imply a negative feedback loop but show no evidence for this. Figure 8 (which is a very good addition to the paper and a nice summary) is rather speculative regarding the relationships of the individual players…

R5: The reviewer's constructive comment is much appreciated as we described in updated Figure. The microRNA profiling analyses showed that shBC200 and OEBC200 contained high and low levels of miR-218-5p, i.e., when BC200 expression was reduced the expression of miR-218-5p was higher and vice-versa, which, gives an indication that miR-218-5p targets /regulate the BC200 expression, the prediction from online databases also confirmed BC200 directly interacts with multiple binding sites to hsa-miR-218-5p. Please see the updated Figure 5C and Figure 5E.

  1. While classical EMT does not occur in GB, as no basement membrane is present and E-cadherin is only rarely expressed in GB (Iwadate, 2016), there is considerable overlap between the gene expression profile of EMT (from human mammary epithelial cells) and (not just the mesenchymal subtype of) GB (Zarkoob et al., 2013), i.e. GB is always invasive! Also Figure 6, proliferation should again be considered.

R6: We appreciate the reviewer’s insightful suggestion. We have made the change as requested by the reviewer. We have included the cell proliferation assay curve comparing the effect of miR-218-5p knockdown, overexpression, and control transfected GBM cells. Please Kindly see our updated Figure 6 and also kindly see in revised results section, line 359-378.

3.6 miR-218-5p modulates stem cell characteristics and TMZ resistance

Prediction binding of miRNAs (miR-218-5p) from DIANA and PITA database shows it targets BC200 RNA [25]. Thus, we wanted to investigate the relevance of miR-218-5p in controlling the self-renewal potential and TMZ resistance of GBM cells. We first verified the expression of miR-218-5p by using qRT-PCR in GBM and control cells (Figure 6A). The expression of miR-218-5p was lower in GBM cells than in normal human astrocytes. Furthermore, the relative expression of miR-218-5p was observed to be negatively correlated with BC200 RNA expression (Figure 6B) in GBM tissues. The self-renewal and pluripotency properties of GBM cells were analyzed by targeting miR-218-5p. The inhibition of miR-218-5p showed a significantly higher tumorsphere and colony-forming abilities (Figure 6C and D) in GBM cells than in the control (NC) mock-transfected groups. Furthermore, the effect of miR-218-5p inhibition was noted on TMZ resistance; miR-218-5p-inhibited cells show higher TMZ resistance than the control mock-transfected groups (Figure 6E) but no significant effect on cell proliferation (Figure 6F). Moreover, markers for self-renewal and ABC Transporters in miR-218-5p-inhibited GBM cells (Figure 6G) were evaluated. Markers such as Oct4, SOX2, KLF4, BCRP1, MDR1, and MRP1 were observed to be upregulated in miR-218-5p-inhibited cells compared with control NC and mock-transfected cells. Results revealed that the protein levels of genes associated with drug resistance, such as MGMT, MLH1, MSH2, MSH6 and PMS2, were significantly higher in the miR-218-5p-inhibited group than in the control NC mock-transfected groups (Figure 6H).

Please also see the updated figure 6 legend, line 380-390.

Figure 6. miR-218-5p regulated cell sphere formation, colony formation, and TMZ resistance in GB cells in vitro. (A) The differential expression of miR-218-5p in GB cell lines and normal human astrocytes. (B) Pearson’s correlation curve identified a negative correlation between BC200 and miR-218-5p in GB tissues. (C) Sphere formation assays showed that inhibition or mock transfection of miR-218-5p regulated GB cell stemness. (D) Colony formation assays showed that inhibition or mock transfection of miR-218-5p regulated GB cell survival. (E) The viability of GB cells with inhibition or mock transfection of miR-218-5p was analyzed through SRB assay 48 h after TMZ (0–1000 μM) treatment. (F) CCK-8 assay showed that inhibition or mock transfection of miR-218-5p had no effect on GB cell proliferation. (G) The protein levels of SOX2, Oct4, BRPC1, MRP1 and MDR1 with inhibition or mock transfection of miR-218-5p. (H) The protein levels of MGMT, MLH1, MSH2, MSH6 and PMS2 following inhibition or mock transfection of miR-218-5p in GB cells were determined through western blot. *P < 0.05, **P < 0.01 and ***P < 0.001.

  1. Figure 7: Tumour wasn't "induced", that term is better reserved for chemical induction and the use of survival curves for subcutaneous models is rather debateable. Also terminating the survival curve at 8 weeks is far too early to make any significant statement regarding survival.

R7: We thank the reviewer for the insightful comment. In updated Figure 7C, we showed the overall survival time of mice treated with the shBC200/TMZ combination compared with that in mice treated with shBC200 and TMZ individually was significantly improved. Please kindly see the revised results section, line 391-409.

3.7 BC200 RNA inactivity sensitized GBM cells to TMZ combined therapy in vivo

We evaluated the potential of BC200 RNA knockdown together with TMZ combination treatment in inhibiting the tumor-initiating ability in xenograft models. Tumor was induced in NOD/SCID mice, which were then treated with shBC200, TMZ, control (Vehicle), and combination (shBC200/TMZ) treatments. Tumor volume, body weight, and survival were analyzed for 28 days after cell implantation. At 28 days, mice were sacrificed for immunohistochemistry and gene expression analysis (qRT-PCR). The combination treatment with shBC200 and TMZ led to a significant reduction in tumor size (Figure 7A). Mice treated with shBC200 and the shBC200/TMZ combination showed increased bodyweight but no sign of toxicity and negative treatment effects (Figure 7B). Moreover, the overall survival time was higher in mice treated with the shBC200/TMZ combination compared with that in mice treated with shBC200 and TMZ individually (Figure 7C). Moreover, comparative real-time PCR analyses showed an increased level of miR-218-5p expression in the shBC200/TMZ combination group in comparison with vehicle, TMZ, and shBC200 groups (Figure 7D), suggesting the role of miR-218-5p in GBM progression. Furthermore, IHC analysis results of tissue sections in GBM patients were consistent with the results of qRT-PCR. IHC results showed that shBC200 and shBC200/TMZ combination groups showed significantly suppressed tumor proliferation, oncogenicity, and tumorigenesis-associated markers (Figure 7E) in tissue samples.

Please also see the updated figure 7 legend, line 411-420.

Figure 7. Combination of shBC200 and TMZ inhibits tumor growth in GB mice models. (A) Tumor size over time curve indicates that the most significantly delayed tumor growth was observed in the shBC200 and TMZ combination group, followed by shBC200 group, whereas TMZ and control groups did not differ significantly. (B) Bodyweight curves over time suggest no clear cytotoxic effects in all mice because no significant decrease in weight was observed. (C) Kaplan–Meier survival curve showed increased median overall survival in TMZ, shBC200, and shBC200/TMZ combination, whereas the combination group showed the best survival ratio. (D) Comparative real-time PCR analyses showed a significantly increased mRNA level of miR-218-5p expression in both shBC200 and shBC200/TMZ combination groups as compared with vehicle and TMZ counterparts. (E) IHC staining showed that treatment in the shBC200 group and shBC200/TMZ combination group suppressed tumorigenesis and TMZ resistance. **P < 0.01, and ***P < 0.001.

  1. The Student's t-test is inappropriate for analysing most data sets presented. While this will not alter the overall message of the paper, it should be addressed.

R8: We thank the reviewer for the comments. We addressed the issue raised in revised statistical analysis section, please see line 226-229.

2.14 Statistical analysis

All experiments were executed in triplicates. The comparison between two groups was done using the 2-sided Student’s t-test, while one-way analysis of variance (ANOVA) was used to compare ≥3 groups. A p-value < 0.05 was considered statistically significant.

Minor points:

  1. You mainly call the tumour (correctly) Glioblastoma, so use GB as abbreviation, see below, Delete "multiforme" from abstract, superfluous. Abstract, line 50: "noncoding RNA BC200/miR-218-5p signaling circuit" format changed Background: GB is a grade IV astrocytoma, there are no Glioblastoma Grade I-III Also reference 2 is out of date, replace with Acta Neuropathologica volume 131, pages803–820(2016) Background, lines 65/66: Recent used too often, inelegant 

R1: We sincerely appreciate the reviewer's insightful comments. As per the suggestions we have addressed the issue raised by the reviewer and made the changes in the main text.

  1. Background, lines 67f: This is phrased as if lncRNAs contribute to GB metastasis, which is an event that is of almost negligible importance in GB, as it occurs in less than two percent. Background, lines 74f: BC200 is expressed "in the presence of neurodegenerative diseases", isn't that rather surprising? These diseases are not characterised by motile cells and if anything, are associated with increased cell death, not resistance to cell death.

R2: We thank the reviewer for the insightful comment. We have reworded the text as per the reviewer's suggestion in our revised manuscript. Kindly see our revised Introduction section, please see line 56-92.

  1. Background

Glioblastoma (GBM) is one of the most common (~30%) and lethal cancers of the central nervous system [1]. The World Health Organization classified Glioma into grades I to IV based on its histopathologically determined malignancy level [2]. GBM, a grade IV astrocytoma, is highly aggressive, malignant and invasive [3]. Despite improved therapeutic interventions, the treatment of high-grade glioma is challenging even after combining radiation therapy with surgical resection [4]. Therefore, the molecular mechanism of GBM pathogenesis and treatment resistance must be explored. With the advances in sequencing technologies, evidence indicates a regulatory role of long noncoding RNA (lncRNA) [5]. Recent studies have shown that lncRNAs play a role in the development, progression, and metastasis [6] of many cancers, including GBM [7]. These lncRNAs consist of >200 nucleotides and cannot translate proteins [8]. Furthermore, studies have reported misregulated lncRNA expressions, which determine the temozolomide (TMZ)-drug sensitivity or TMZ-drug resistance of GBM [9]. This lncRNA exerts its functions through lncRNA–miRNA interactions and mRNA silencing [10].

BCYRN1, also known as brain cytoplasmic 200 (BC200) RNA, is a lncRNA highly expressed in the brain (neuron-specific transcript) in the presence of neurodegenerative diseases [11]. Sometimes, BC200 RNA expression is atypically elevated in the presence of various human cancers [12] and its expression is higher in invasive cancer cells than in benign tumors [13]. BC200 RNA was also find in blood specimens such as breast cancer and hepatocellular carcinoma [14]. These data suggest the BC200 RNA has potential value as a diagnostic and prognostic marker for human cancers. However, information regarding BC200 RNA as a clinical biomarker or therapeutic target for GBM is still lacking. The exact underlying molecular mechanism of BC200 RNA in glioma development and progression is still under investigation.

In this study, the role of BC200 RNA in GBM was evaluated to understand the molecular mechanism of chemoresistance and disease progression. The expression level of BC200 RNA in the tissue of GBM patients and GBM cell lines was evaluated. The SRB assay was used to determine the effect of TMZ treatment on GB cells. Furthermore, the stem cell population was evaluated through flow cytometric analysis of GBM cell lines. Overexpression and silencing of BC200 RNA showed its role in survival and GBM tumorigenesis reduction both in vitro and in vivo. Ours is the first study to show the molecular mechanism of BC200 RNA in the promotion of TMZ resistance in GBM through miR-218-5p expression modulation. Thus, BC200 RNA is suggested as a potential clinical biomarker or therapeutic target for GBM.

  1. Why was RPMI medium used for the cell lines, ATCC recommends DMEM, Results, line 214: GB patients and normal controls, Results, line 238: "the likely GBM cells" unclear. Results, line 245: "significantly highly" is ungrammatical and unclear, expressed higher in two of four GB cells.

R3: We sincerely appreciate the reviewer's comments; we addressed the issue raised in revised material and methods section, please see line 102-109.

2.2 Cell lines and cell culture

Human GB likely cell lines T98G (ATCC® CRL-1690™) (ATCC, Manassas, VA, USA) and U87MG (ATCC® HTB-14™) (ATCC, Manassas, VA, USA) were cultured in DMEM; DBTRG-05MG (ATCC® CRL-2020™) (ATCC, Manassas, VA, USA) and GB cells line GBM8901 (Bioresource Collection and Research Center, Taiwan) were cultured in RPMI, supplemented with streptomycin (100 μg/mL), penicillin (100 IU/mL), and 10% fetal bovine serum as a monolayer in a humidified 5% CO2 atmosphere at 37°C and were subcultured every 2–3 days.

  1. Figure 3: If you define wound closure of controls as 100% you should not have any StDev on the control bar, ditto for invasion and colony forming assays. Results, line 293: significantly high(er)

R4: The reviewer's constructive comment is much appreciated. We have reworded the text as per the reviewer's suggestion. Please see line 226-229.

2.14 Statistical analysis

All experiments were executed in triplicates. The comparison between two groups was done using the 2-sided Student’s t-test, while one-way analysis of variance (ANOVA) was used to compare ≥3 groups. A p-value < 0.05 was considered statistically significant.

  1. Discussion, line 402: "Studies have shown that lncRNAs are widely accepted" unclear, are author arguing that the existence of lncRNA is no longer debated? Wasn't aware that was ever the case.

R5: We thank the reviewer for the comments. We addressed the issue raised in our discussion section, please see line 423-443.

Despite major developments in therapeutic interventions, GBM is one of the most lethal and invasive malignancies of the central nervous system and is the leading cause of overall cancer-associated mortality. Patients with GBM exhibit poor prognosis [26] and the current therapy mostly fails because these patients develop multidrug resistance and do not respond to the treatment [27, 28, 29]. Studies have shown that lncRNAs are widely accepted and play a crucial role in the development, progression, and metastasis of cancer as well as in drug resistance of GBM. The expression of BC200, a lncRNA, is high in the neuron-specific transcript in the brain [11]. The increased expression of BC200 RNA was atypically found in various human cancers, and its expression was higher in invasive cancers than in benign tumors [13, 14]. However, BC200 RNA as a clinical therapeutic target or biomarker for GBM is still understudied. In this study, the molecular mechanism of BC200 RNA in glioma development and progression was explored.

    Regulatory BC200 RNA is highly expressed in the peripheral blood of patients with breast cancer [30], suggesting that it can be an important molecular tumor biomarker for indicating human malignancies [31]. In our study, we demonstrated that BC200, both at protein and mRNA level, was significantly upregulated in blood and tissue samples of GBM patients (Figure 1). The expression of Ki-67 and p53-mut markers are indicative of prognosis and survival of GBM patients [17, 18], indicating a positive and significant correlation of BC200 RNA expression with Ki-67and p53. Thus, BC200 RNA is strongly related to poor prognosis of GBM patients.

  1. Discussion, line 420ff: Both, the existence of stem cells in GB and their role in TMZ-resistance are not as straight forward as with other tumours and should be discussed a bit more.

R6: The reviewer's constructive comment is much appreciated. We have added more about the therapy resistance in the revised discussion section. Please kindly see line 445-456.

CSCs are now increasingly being recognized as a critical target in cancer treatment because these stem cells are often correlated with the resistance against conventional chemo- and radiotherapy. Overcoming their treatment resistance is the key issue in cancer therapeutics [32, 33]. Moreover, studies have reported that misregulated lncRNA expression remains associated with the TMZ drug resistance of GBM. In this study, we observed that the expression of BC200 increased in GBM cells compared with normal brain cells (Figure 2). Furthermore, this study showed that GBM cells exhibited high expression of ALDH+/CD133+ cells, and key markers associated with drug resistance, proliferation, invasions and metastasis, such as MGMT, MLH1, MSH2, MSH6, PMS2, BCRP1, MDR1, and MRP1, were associated with chemoresistance [9]. Moreover, we found that in GBM cells, the expression of these markers together with BC200 RNA expression is significantly higher, suggesting their role in TMZ chemoresistance.

  1. Misregulated or dysregulated?

R7: We have rechecked and amended the changes in the text. Please see line 445-456.

CSCs are now increasingly being recognized as a critical target in cancer treatment because these stem cells are often correlated with the resistance against conventional chemo- and radiotherapy. Overcoming their treatment resistance is the key issue in cancer therapeutics [32, 33]. Moreover, studies have reported that dysregulated lncRNA expression remains associated with the TMZ drug resistance of GBM. In this study, we observed that the expression of BC200 increased in GBM cells compared with normal brain cells (Figure 2). Furthermore, this study showed that GBM cells exhibited high expression of ALDH+/CD133+ cells, and key markers associated with drug resistance, proliferation, invasions and metastasis, such as MGMT, MLH1, MSH2, MSH6, PMS2, BCRP1, MDR1, and MRP1, were associated with chemoresistance [9]. Moreover, we found that in GBM cells, the expression of these markers together with BC200 RNA expression is significantly higher, suggesting their role in TMZ chemoresistance.

  1. Discussion, line 430f: significant high is not a good term, see other comments regarding this.

R8: We thank the reviewer for their insightful comments. We addressed the issue raised in our manuscript and rechecked the entire text to improve it.

  1. Discussion, line 443: (Figure 5A) should be bold

R9: We have rechecked and amended the required changes in the text. Please see line 466-469.

Additionally, to understand the underlying molecular mechanism of BC200 in GBM chemosensitivity, a cell viability assay was performed. BC200-silenced cells showed decreased resistance to TMZ compared with the control group (Figure 5A), whereas OEBC200 showed the reverse effect (Figure 5).

  1. Lines 494ff: Font is wrong for some names

R10: We thank the reviewer for the comments. We addressed the issue raised in our revised manuscript. Please see line 488-496.

Finally, we demonstrated that the combination of BC200 inhibition and TMZ treatment increased GBM sensitivity. It greatly inhibited the tumor growth in the xenograft mouse model with no negative impact on body weight and survival (Figure 7). The results of IHC analysis are concordant with those of qRT-PCR; the shBC200/TMZ combination group shows significant suppression of tumor proliferation, oncogenicity, and tumorigenesis-associated markers. Moreover, the miR-218-5p level expressed was significantly high in the combination treatment group. Collectively, we provided a strong preclinical evidence in support of using shBC200/TMZ combination for treating malignant glioma cancer. In addition, miR-218-5p could be used as a biomarker for monitoring therapeutic responses.

Reviewer 2 Report

In this manuscript, Su et al report on the impact of a specific non-coding RNA, known as BC200, on glioblastoma growth, invasion, self-removal, sensitivity to temozolomide (TMZ) and other endpoints. They present a lot of data, but a clear working hypothesis is lacking and the interpretation of some oft the data is misleading or even wrong.

1) Thus, they observed BC200 RNA at elevated level in lymphocytes of GBM patients, although lymphocytes of patients are normal cells. They do not give an explanation why in PBMC of cancer patients this RNA fraction is enhanced (about 2-fold).

2) They compared U87MG (presumably an astrocytoma grade III line) with T98G (a gdare IV glioblastoma line). U87MG is more sensitive than T98G (Fig. 2), which is well known, because of high MGMT expression in T98G (see Hermisson et al., J. Neurochem., 96, 788-776, 2006). This important point is completely neglected. ALDH goes parallel to resistance, but it is not the cause of TMZ resistance. ALDH is neither involved in DNA repair nor DNA damage response nor checkpoint activation and apoptosis execution. The role of ALDH in TMZ resistance is still unknown.

3) It is well known that the key mechanisms determining TMZ resistance are low mismatch repair (MSH2, MSH6, MLH1, PMS2), high MGMT and high DSB repair, notably by upregulation of Rad51, BRCA1, XRCC3 and others. It is also known that apoptosis execution might play a role, but this is less frequent happening than MGMT downregulation, which is silenced due to promoter methylation in ~ 40% of GBM. These points are completely neglected in this study. It would be worth examining the effect of up- and downregulation of BC200 on MMR, MGMT and homologous recombination repair activity. Please note that U87MG is MGMT promoter methylated and exhibits low mismatch repair gene expression (see Hermisson et al, 2006), which makes this strain not responding well to TMZ than other MGMT lacking GBM cell lines.

4) The manuscript is full of data, but it seems to me that much of it are not relevant for TMZ. In Fig. 8, the data and conclusions are compiled and indicate the way of misinterpretation. They suggest miR-218 regulates ABC transporter and, therefore, TMZ resistance /sensitivity. But TMZ and the decomposition product diazomethane is not subject of transport by ABC transporter. These are small molecules that simply penetrate cells and methylate DNA and other cellular targets, including mitochondrial DNA. The toxic effect brought about by specific DNA lesions is dependent on cell growth, i.e., proliferation plays an enormous role in TMZ resistance and cells having a long cell cycle or arrested in proliferation show a low or even no response to clinically relevant doses of TMZ. Therefore, any change in growth behaviour by regulation of stem cells factors will have an impact on TMZ resistance. If cells are not proliferating, they are completely resistant to TMZ. Thus, proliferation capacity must be taken into account as well as MGMT, MMR and HR, if the effect of any player on TMZ resistance is properly explained.

Author Response

Point-by-point responses to reviewer’s comments:

We would like to thank all the reviewers for the thorough reading of our manuscript as well as their valuable comments. We have followed the reviewer’s comments thoroughly and feel that they have further helped in strengthening the manuscript.

Reviewer 2 Comments and Suggestions for Authors

In this manuscript, Su et al report on the impact of a specific non-coding RNA, known as BC200, on glioblastoma growth, invasion, self-removal, sensitivity to temozolomide (TMZ) and other endpoints. They present a lot of data, but a clear working hypothesis is lacking and the interpretation of some of the data is misleading or even wrong.

  1. Thus, they observed BC200 RNA at elevated level in lymphocytes of GBM patients, although lymphocytes of patients are normal cells. They do not give an explanation why in PBMC of cancer patients this RNA fraction is enhanced (about 2-fold).

R1: We thank the reviewer for the comments and apologize for the confusion and our oversight. In updated Figure 1, we are evaluating the expression level of BC200 in GBM and normal patients. For that the we initially detected the expression level of BC200 RNA in the blood (Whole blood and Peripheral blood; PBMC) of GBM patients and normal controls. Results revealed elevated levels (~2 folds) of BC200 RNA expression in GBM patients compared with healthy participants, and BC200 RNA was also find in blood specimens such as breast cancer and hepatocellular carcinoma. Pleased kindly see our revised manuscript, please see line 72-81.

BCYRN1, also known as brain cytoplasmic 200 (BC200) RNA, is a lncRNA highly expressed in the brain (neuron-specific transcript) in the presence of neurodegenerative diseases [11]. Sometimes, BC200 RNA expression is atypically elevated in the presence of various human cancers [12] and its expression is higher in invasive cancer cells than in benign tumors [13]. BC200 RNA was also find in blood specimens such as breast cancer and hepatocellular carcinoma [14]. These data suggest the BC200 RNA has potential value as a diagnostic and prognostic marker for human cancers. However, information regarding BC200 RNA as a clinical biomarker or therapeutic target for GBM is still lacking. The exact underlying molecular mechanism of BC200 RNA in glioma development and progression is still under investigation.

  1. They compared U87MG (presumably an astrocytoma grade III line) with T98G (a gdare IV glioblastoma line). U87MG is more sensitive than T98G (Fig. 2), which is well known, because of high MGMT expression in T98G (see Hermisson et al., J. Neurochem., 96, 788-776, 2006). This important point is completely neglected. ALDH goes parallel to resistance, but it is not the cause of TMZ resistance. ALDH is neither involved in DNA repair nor DNA damage response nor checkpoint activation and apoptosis execution. The role of ALDH in TMZ resistance is still unknown.

R2: Reviewer's comments are greatly appreciated. We have included the expression of MGMT in updated Figure 2D and Figure 5B. Please kindly see revised results section, line 253-269 and line 331-350.

3.2 High BC200 RNA expression associated with TMZ resistance of stem-cell-like population in GBM cells

We attempted to confirm the role of BC200 RNA expression in chemo-resistance through upregulation of the stem-cell-like population in GBM. We first evaluated the differential expression of BC200 RNA in GBM (U87MG, DBTRG-05MG, T98G, and GBM8901) and normal human astrocytes (NHA) cells. BC200 RNA expression was significantly higher in GBM cell lines than in NHA cells (Figure 2A). The evaluation of cell viability after TMZ treatment revealed that GBM cells showed higher chemo-resistance (TMZ) than the likely GBM cells (Figure 2B). Previous studies have suggested that the increased expression of ALDH+ and CD133+ cells was associated with mesenchymal phenotype and chemoresistance [19], cancer recurrence, and poor prognosis [20]. Furthermore, the role of BC200 RNA in TMZ chemoresistance was assessed through flow cytometric analysis, revealing a significantly higher number of ALDH+/CD133+ cells in cancer cells than in normal human astrocytes (Figure 2C). The expression levels of key markers associated with drug resistance [21, 22, 23], proliferation, migration, and invasion, such as BCRP1, MDR1, MRP1 and MGMT along with BC200 RNA, were significantly highly expressed in GBM cells (Figure 2D).

3.5 BC200 RNA expression associated with TMZ chemoresistance and miR-218-5p expression

We demonstrated that BC200 RNA expression plays a crucial role in TMZ resistance. To further understand the underlying mechanism of BC200 RNA in GBM chemosensitivity, cell viability assay was performed on GBM cells with silenced and overexpressed BC200 RNA. The results revealed that the silenced BC200 (shBC200) cells showed decreased TMZ resistance compared with the control group, and OEBC200 cells showed increased TMZ chemoresistance (Figure 5A). Furthermore, we evaluated the expression level of markers associated with drug resistance, proliferation, migration, and invasion in GBM cells. Western blot analysis results showed that the levels of MGMT (O-6-Methylguanine-DNA Methyltransferase), BCRP1 (ABCG2 human, ABC transporter), MDR1 (Multidrug Resistance Protein), and MRP1 proteins were significantly higher in OEBC200 cells than in BC200-silenced cells (Figure 5B). To further correlate the effect of BC200 RNA on miRNA expression, we applied bioinformatic analysis and observed that miR expression was inversely correlated with BC200 RNA expression. As shown in the heatmap of shBC200 and OEBC200 groups, the level of miR-218-5p was negatively associated with BC200 RNA expression (Figure 5C). The binding prediction of these miRNAs (miR-218-5p) from DIANA and PITA database [25] shows that BC200 RNA is targeted by miR-218-5p (Figure 5D and E). This indicates that a key interaction exists between BC200 RNA and miR-218-5p.

Please also see the updated figure 2 legend, line 271-275.

Figure 2. Differential expression of BC200 RNA in GBM cell lines. (A) The differential expression of BC200 RNA in GBM cell lines and normal human astrocytes are shown. (B) The viability of GBM cells was analyzed through SRB assay 48 h after TMZ (0–1000 μM) treatment. (C) Flow cytometry analysis of the ALDH1+/CD133+ portion in GBM cell lines and normal human astrocytes. (D) The level of BC200, BCRP1, MDR1, MRP1 and MGMT in GBM cell lines was analyzed using qRT-PCR. *P < 0.05, **P < 0.01, and ***P < 0.001.

Please also see the updated figure 5 legend, line 352-358.

Figure 5. BC200 RNA promotes TMZ resistance in GBM through sponge miR-218-5p. (A) The viability of shBC200 and OEBC200 GBM cell lines was analyzed through SRB assay 48 h after TMZ (0–1000 μM) treatment. (B) The levels of MGMT, BCRP1, MDR1, and MRP1 following shBC200 and OEBC200 in GBM cells were determined through western blot. (C) MicroRNA profiling analyses showed that shBC200 and OEBC200 contained high and low levels of miR-218-5p, respectively. (D) LncBase Predicted v.2 predicted that a high binding score of miR-218-5p with BC200. (E) BC200 directly interacts with multiple binding sites to hsa-miR-218-5p.

  1. It is well known that the key mechanisms determining TMZ resistance are low mismatch repair (MSH2, MSH6, MLH1, PMS2), high MGMT and high DSB repair, notably by upregulation of Rad51, BRCA1, XRCC3 and others. It is also known that apoptosis execution might play a role, but this is less frequent happening than MGMT downregulation, which is silenced due to promoter methylation in ~ 40% of GBM. These points are completely neglected in this study. It would be worth examining the effect of up- and downregulation of BC200 on MMR, MGMT and homologous recombination repair activity. Please note that U87MG is MGMT promoter methylated and exhibits low mismatch repair gene expression (see Hermisson et al, 2006), which makes this strain not responding well to TMZ than other MGMT lacking GBM cell lines.

R3: The reviewer’s constructive comment is greatly appreciated. We have included MMR, MGMT and homologous recombination repair activity in updated Figure 2D Figure 5B and Figure 6H. Please also kindly see revised results section, please see line 359-378.

3.6 miR-218-5p modulates stem cell characteristics and TMZ resistance

Prediction binding of miRNAs (miR-218-5p) from DIANA and PITA database shows it targets BC200 RNA [25]. Thus, we wanted to investigate the relevance of miR-218-5p in controlling the self-renewal potential and TMZ resistance of GBM cells. We first verified the expression of miR-218-5p by using qRT-PCR in GBM and control cells (Figure 6A). The expression of miR-218-5p was lower in GBM cells than in normal human astrocytes. Furthermore, the relative expression of miR-218-5p was observed to be negatively correlated with BC200 RNA expression (Figure 6B) in GBM tissues. The self-renewal and pluripotency properties of GBM cells were analyzed by targeting miR-218-5p. The inhibition of miR-218-5p showed a significantly higher tumorsphere and colony-forming abilities (Figure 6C and D) in GBM cells than in the control (NC) mock-transfected groups. Furthermore, the effect of miR-218-5p inhibition was noted on TMZ resistance; miR-218-5p-inhibited cells show higher TMZ resistance than the control mock-transfected groups (Figure 6E) but no significant effect on cell proliferation (Figure 6F). Moreover, markers for self-renewal and ABC Transporters in miR-218-5p-inhibited GBM cells (Figure 6G) were evaluated. Markers such as Oct4, SOX2, KLF4, BCRP1, MDR1, and MRP1 were observed to be upregulated in miR-218-5p-inhibited cells compared with control NC and mock-transfected cells. Results revealed that the protein levels of genes associated with Drug resistance, such as MGMT, MLH1, MSH2, MSH6 and PMS2, were significantly higher in the miR-218-5p-inhibited group than in the control NC mock-transfected groups (Figure 6H).

Please also see the updated figure 6 legend, line 380-390.

Figure 6. miR-218-5p regulated cell sphere formation, colony formation, and TMZ resistance in GB cells in vitro. (A) The differential expression of miR-218-5p in GB cell lines and normal human astrocytes. (B) Pearson’s correlation curve identified a negative correlation between BC200 and miR-218-5p in GB tissues. (C) Sphere formation assays showed that inhibition or mock transfection of miR-218-5p regulated GB cell stemness. (D) Colony formation assays showed that inhibition or mock transfection of miR-218-5p regulated GB cell survival. (E) The viability of GB cells with inhibition or mock transfection of miR-218-5p was analyzed through SRB assay 48 h after TMZ (0–1000 μM) treatment. (F) CCK-8 assay showed that inhibition or mock transfection of miR-218-5p had no effect on GB cell proliferation. (G) The protein levels of SOX2, Oct4, BRPC1, MRP1 and MDR1 with inhibition or mock transfection of miR-218-5p. (H) The protein levels of MGMT, MLH1, MSH2, MSH6 and PMS2 following inhibition or mock transfection of miR-218-5p in GB cells were determined through western blot. *P < 0.05, **P < 0.01 and ***P < 0.001.

  1. The manuscript is full of data, but it seems to me that much of it are not relevant for TMZ. In Fig. 8, the data and conclusions are compiled and indicate the way of misinterpretation. They suggest miR-218 regulates ABC transporter and, therefore, TMZ resistance /sensitivity. But TMZ and the decomposition product diazomethane is not subject of transport by ABC transporter. These are small molecules that simply penetrate cells and methylate DNA and other cellular targets, including mitochondrial DNA. The toxic effect brought about by specific DNA lesions is dependent on cell growth, i.e., proliferation plays an enormous role in TMZ resistance and cells having a long cell cycle or arrested in proliferation show a low or even no response to clinically relevant doses of TMZ. Therefore, any change in growth behaviour by regulation of stem cells factors will have an impact on TMZ resistance. If cells are not proliferating, they are completely resistant to TMZ. Thus, proliferation capacity must be taken into account as well as MGMT, MMR and HR, if the effect of any player on TMZ resistance is properly explained.

R4: Reviewer’s comments are greatly appreciated. We have included the proliferation capacity of shBC200/OEBC200 transfected GBM cells updated Figure 3F and Figure 4F.

We have included MMR, MGMT and homologous recombination repair activity in updated Figure 2D Figure 5B and Figure 6H.

In addition, we have redrawn the updated Figure 8.

Reviewer 3 Report

This manuscript reported a study that BC200 is overexpressed in glioblastoma (GBM) and contributes to its resistance to Temozolomide (TMZ). BC200 was shown to regulate stem cell factors and proteins involved with drug efflux. In TMZ-resistant GBM, this study also showed BC200 level is negatively correlated with miR-218-5p. Attenuation of BC200 in the GBM mouse model was found to inhibit tumor progression. Although the findings are novel, several issues need to be clarified and are discussed in the following.

  1. Dot plots should be used in Fig. 1B
  2. Does the Clone Do-7 antibody detect only p53-mutant but not the p53-wt? Why p53-mut in Figs. 1C-D?
  3. Please define which cell lines are likely GBM cells in Fig. 2B.
  4. The significance indexes (stars) in Fig. 2D were reference to which cell line?
  5. Why two different cell lines were used between BC200 knocked down and overexpression experiments? The same cell lines should be used for comparison.
  6. Does negative correlation between BC200 and miR-218-5p mean that miR-218-5p regulates BC200 expression or targets BC200 degradation?
  7. References for miR-218-5p on line 337 should be provided.
  8. What was used to inhibit miR-218-5p?
  9. How was shBC200 delivered to the tumor in the mouse model? How did shBC200 enter the tumor cells in mice?
  10. Why was the IHC analysis performed in GBM patients in Fig. 7E? Is this study performed using GBM patients?

Author Response

Point-by-point responses to reviewer’s comments:

We would like to thank all the reviewers for the thorough reading of our manuscript as well as their valuable comments. We have followed the reviewer’s comments thoroughly and feel that they have further helped in strengthening the manuscript.

Reviewer 3 Comments and Suggestions for Authors

This manuscript reported a study that BC200 is overexpressed in glioblastoma (GBM) and contributes to its resistance to Temozolomide (TMZ). BC200 was shown to regulate stem cell factors and proteins involved with drug efflux. In TMZ-resistant GBM, this study also showed BC200 level is negatively correlated with miR-218-5p. Attenuation of BC200 in the GBM mouse model was found to inhibit tumor progression. Although the findings are novel, several issues need to be clarified and are discussed in the following.

  1. Dot plots should be used in Fig. 1B

R1: We thank the reviewer for the suggestion. The Figure 1B has been revised accordingly, please kindly see our updated Figure 1B.

Please also see the updated figure 1 legend, line 248-252.

Figure 1. Expression level of full-length BC200 RNA in the blood and tissue. (A) Whole blood or PBMC from healthy participants and GBM patients were assessed using qRT-PCR. GAPDH was used as the endogenous control. (B) The differential expression of BC200 in GBM samples (n = 48) and adjacent normal tissues (n = 15) is shown. (C and D) The differential expression of Ki-67 and p53-mut in GBM patients with a high or low expression of BC200 RNA. *P < 0.05, **P < 0.01, and ***P < 0.001.

  1. Does the Clone Do-7 antibody detect only p53-mutant but not the p53-wt? Why p53-mut in Figs. 1C-D?

R2: We thank the reviewer for the comments; we apologized for the typing error. In Fig 1C, the IHC staining was using the mut-p53 (clone Ab-3, Merck, Rd, Wilson, USA) antibody. In updated Figure 1C-D, the correlation analysis of the expression of BC200 with Ki-67 and p53-mut has been shown. Please see line 234-246.

3.1 Upregulation of BC200 RNA expression in GBM patients

To evaluate the role of BC200 RNA in GBM, we initially detected the expression level of BC200 RNA in the blood (whole blood and peripheral blood) of GBM and normal patients. Results revealed elevated levels (~2 folds) of BC200 RNA expression in GBM patients compared with healthy participants (Figure 1A). Furthermore, the results of IHC analysis of tissue sections obtained from GBM patients showed concordance with those of qRT-PCR. Quantitative analysis shows that BC200 RNA expression was higher in 48 GBM tissues than in 15 normal adjacent tissues (Figure 1B). The widely used markers to determine the prognostic behavior of brain tumors and survival of GBM patients are Ki-67 and p53 [17, 18]. Figure 1C and 1D indicates that high BC200 RNA expression was positively correlated with Ki-67 (p<0.001) and p53-mut expression (p<0.01), suggesting that BC200 RNA is strongly associated with unfavorable prognoses in GBM patients.

Please also see the updated figure 1 legend, line 248-252.

Figure 1. Expression level of full-length BC200 RNA in the blood and tissue. (A) Whole blood or PBMC from healthy participants and GBM patients were assessed using qRT-PCR. GAPDH was used as the endogenous control. (B) The differential expression of BC200 in GBM samples (n = 48) and adjacent normal tissues (n = 15) is shown. (C and D) The differential expression of Ki-67 and p53-mut in GBM patients with a high or low expression of BC200 RNA. *P < 0.05, **P < 0.01, and ***P < 0.001.

  1. Please define which cell lines are likely GBM cells in Fig. 2B.

R3: We thank the reviewer for the comments. We have incorporated this information in the revised materials and methods section, please see line 102-109.

2.2 Cell lines and cell culture

Human GB likely cell lines T98G (ATCC® CRL-1690™) (ATCC, Manassas, VA, USA) and U87MG (ATCC® HTB-14™) (ATCC, Manassas, VA, USA) were cultured in DMEM; DBTRG-05MG (ATCC® CRL-2020™) (ATCC, Manassas, VA, USA) and GB cells line GBM8901 (Bioresource Collection and Research Center, Taiwan) were cultured in RPMI, supplemented with streptomycin (100 μg/mL), penicillin (100 IU/mL), and 10% fetal bovine serum as a monolayer in a humidified 5% CO2 atmosphere at 37°C and were sub-cultured every 2–3 days.

  1. The significance indexes (stars) in Fig. 2D were reference to which cell line?

R4: We thank the reviewers for their comments. The U87MG and DBTRG05MG group were used as the reference cell line for comparing the expression and showing the reference. We have improved updated Figure 2D and the corresponding result section of our manuscript text. Please see line 253-269.

3.2 High BC200 RNA expression associated with TMZ resistance of stem-cell-like population in GBM cells

We attempted to confirm the role of BC200 RNA expression in chemo-resistance through upregulation of the stem-cell-like population in GBM. We first evaluated the differential expression of BC200 RNA in GBM (U87MG, DBTRG-05MG, T98G, and GBM8901) and normal human astrocytes (NHA) cells. BC200 RNA expression was significantly higher in GBM cell lines than in NHA cells (Figure 2A). The evaluation of cell viability after TMZ treatment revealed that GBM cells showed higher chemo-resistance (TMZ) than the likely GBM cells (Figure 2B). Previous studies have suggested that the increased expression of ALDH+ and CD133+ cells was associated with mesenchymal phenotype and chemoresistance [19], cancer recurrence, and poor prognosis [20]. Furthermore, the role of BC200 RNA in TMZ chemoresistance was assessed through flow cytometric analysis, revealing a significantly higher number of ALDH+/CD133+ cells in cancer cells than in normal human astrocytes (Figure 2C). The expression levels of key markers associated with drug resistance [21, 22, 23], proliferation, migration, and invasion, such as BCRP1, MDR1, MRP1 and MGMT along with BC200 RNA, were significantly highly expressed in GBM cells (Figure 2D).

Please also see the updated figure 2 legend, line 271-275.

Figure 2. Differential expression of BC200 RNA in GBM cell lines. (A) The differential expression of BC200 RNA in GBM cell lines and normal human astrocytes are shown. (B) The viability of GBM cells was analyzed through SRB assay 48 h after TMZ (0–1000 μM) treatment. (C) Flow cytometry analysis of the ALDH1+/CD133+ portion in GBM cell lines and normal human astrocytes. (D) The level of BC200, BCRP1, MDR1, MRP1 and MGMT in GBM cell lines was analyzed using qRT-PCR. *P < 0.05, **P < 0.01, and ***P < 0.001.

  1. Why two different cell lines were used between BC200 knocked down and overexpression experiments? The same cell lines should be used for comparison.

R5: We thank the reviewer for the insightful comment. The relative expression of BC200 in GBM cells are different; the expression of BC200 RNA in GB cells (U87MG, DBTRG-05MG) was less. That’s why we used it for over-expression experiments, and its expression in GBM cells (T98G, and GBM8901) are comparatively higher than the U87MG, DBTRG-05MG cells, that’s why we used them for knockdown experiments.

  1. Does negative correlation between BC200 and miR-218-5p mean that miR-218-5p regulates BC200 expression or targets BC200 degradation?

R6: The reviewer’s constructive comment is much appreciated. A correlation study was mainly used to predict the target of miR’s and genes. A strong negative correlation was observed when genes were downregulated, and miR was upregulated. As described in updated Figure 5C. The microRNA profiling analyses showed that shBC200 and OEBC200 contained high and low levels of miR-218-5p, i.e., when BC200 expression was reduced the expression of miR-218-5p was higher and vice-versa, which, gives an indication that miR-218-5p targets /regulate the BC200 expression.   

  1. References for miR-218-5p on line 337 should be provided. What was used to inhibit miR-218-5p?

R7: The reviewer’s comments are greatly appreciated. We have incorporated the details of miR-218-5p inhibitors and mimics used for inhibiting the miR-218-5p in the revised materials and methods section, please see line 129-138.

2.5 Vector construction and infection

Lentivirus containing BC200 short hairpin (shBC200) RNA and BC200 overexpression (OEBC200) vectors were purchased from ThermoFisher Scientific (USA) and were used acording to the manufacturer’s instructions. Two clones of shRNA were used to effectively knockdown (shBC200) and overexpress (OEBC200) BC200; the complete procedure of shRNA lentivirus infection and construction was conducted according to the practice guidelines at a certified BSL-2 laboratory, The Integrated Laboratories for Translational Medicine, Taipei Medical University. Human GB cells were transfected with miR-218 (mimic), miR-negative control (miR-NC), and miR-218 (inhibitor) were purchased from Qiagen and used as per the manufacturer’s protocol.

  1. How was shBC200 delivered to the tumor in the mouse model? How did shBC200 enter the tumor cells in mice?

R8: We thank the reviewer for the insightful comment. shRNA (shBC200) plasmid was used for transfecting and generating stable BC200-silenced GBM8901 clones, and these BC200-silenced GBM8901 cells were grown into tumor spheres and were injected into the mice. We have incorporated this information in the materials and methods section where vector construction and infection protocols and in-vivo studies were described in revised material and method section, please see line 211-225.

2.13 In vivo studies

An animal study was conducted according to the protocols approved by the Taipei Medical University (Laboratory Animal Center, Affidavit of Approval of Animal Use Protocol, Taipei Medical University, protocol no. LAC-2017-0512). NOD/SCID mice (6–8 weeks old) were purchased from BioLASCO Taiwan Co., Ltd. A subcutaneous GBM xenograft mouse model was established using tumor spheres grown from GBM8901 cells (1 × 106 cells/20 µL/injection). All treatments started when the tumor became palpable. Mice were then randomly subdivided into four groups: vehicle control (sham injection), TMZ (0.9 mg/kg, p.o., five times/week), shBC200 (where BC200-silenced GBM8901 tumor spheres were injected), and combination of shBC200 with TMZ treatment (0.9 mg/kg, p.o., five times/week). The tumor size was measured using a standard caliper once a week and expressed in cubic centimeters using the formula tumor volume = (length × width2)/2, where length represents the longest tumor diameter, and width represents the perpendicular tumor diameter. After the experimental period, mice were sacrificed humanely and tumor samples were collected for further analysis.

  1. Why was the IHC analysis performed in GBM patients in Fig. 7E? Is this study performed using GBM patients?

R9: The reviewer’s comments are greatly appreciated. The in-vivo experiments of this study were performed using mice tumor samples. We have improved updated Figure 7E and the corresponding result section of our manuscript text. Please see line 391-409.

3.7 BC200 RNA inactivity sensitized GBM cells to TMZ combined therapy in vivo

We evaluated the potential of BC200 RNA knockdown together with TMZ combination treatment in inhibiting the tumor-initiating ability in xenograft models. Tumor was induced in NOD/SCID mice, which were then treated with shBC200, TMZ, control (Vehicle), and combination (shBC200/TMZ) treatments. Tumor volume, body weight, and survival were analyzed for 28 days after cell implantation. At 28 days, mice were sacrificed for immunohistochemistry and gene expression analysis (qRT-PCR). The combination treatment with shBC200 and TMZ led to a significant reduction in tumor size (Figure 7A). Mice treated with shBC200 and the shBC200/TMZ combination showed increased bodyweight but no sign of toxicity and negative treatment effects (Figure 7B). Moreover, the overall survival time was higher in mice treated with the shBC200/TMZ combination compared with that in mice treated with shBC200 and TMZ individually (Figure 7C). Moreover, comparative real-time PCR analyses showed an increased level of miR-218-5p expression in the shBC200/TMZ combination group in comparison with vehicle, TMZ, and shBC200 groups (Figure 7D), suggesting the role of miR-218-5p in GBM progression. Furthermore, IHC analysis results of tissue sections in GBM patients were consistent with the results of qRT-PCR. IHC results showed that shBC200 and shBC200/TMZ combination groups showed significantly suppressed tumor proliferation, oncogenicity, and tumorigenesis-associated markers (Figure 7E) in tissue samples.

Please also see the updated figure 7 legend, line 411-420.

Figure 7. Combination of shBC200 and TMZ inhibits tumor growth in GB mice models. (A) Tumor size over time curve indicates that the most significantly delayed tumor growth was observed in the shBC200 and TMZ combination group, followed by shBC200 group, whereas TMZ and control groups did not differ significantly. (B) Bodyweight curves over time suggest no clear cytotoxic effects in all mice because no significant decrease in weight was observed. (C) Kaplan–Meier survival curve showed increased median overall survival in TMZ, shBC200, and shBC200/TMZ combination, whereas the combination group showed the best survival ratio. (D) Comparative real-time PCR analyses showed a significantly increased mRNA level of miR-218-5p expression in both shBC200 and shBC200/TMZ combination groups as compared with vehicle and TMZ counterparts. (E) IHC staining showed that treatment in the shBC200 group and shBC200/TMZ combination group suppressed tumorigenesis and TMZ resistance. **P < 0.01, and ***P < 0.001.

Round 2

Reviewer 1 Report

My concerns were all adressed, thank you for your Revision. Could you just check following sentence, as it doesn't make sense/isn't clear:

"The evaluation of cell viability after TMZ treatment revealed that GBM cells showed higher chemo-resistance (TMZ) than the likely GBM cells (Figure 2B)."

Author Response

Point-by-point responses to reviewer's comments:

We would like to thank all the reviewers for the thorough reading of our manuscript as well as their valuable comments. We have followed the reviewer's comments thoroughly and feel that they have further helped in strengthening the manuscript.

Reviewer 1 Comments and Suggestions for Authors

My concerns were all adressed, thank you for your Revision. Could you just check following sentence, as it doesn't make sense/isn't clear? "The evaluation of cell viability after TMZ treatment revealed that GBM cells showed higher chemo-resistance (TMZ) than the likely GBM cells (Figure 2B)."

R1: We thank the reviewer for the insightful comment and apologize for the confusion. We have reworded the text in our revised manuscript. Kindly see line 261-278.

3.2 High BC200 RNA expression associated with TMZ resistance of stem-cell-like population in GBM cells

We attempted to confirm the role of BC200 RNA expression in chemo-resistance through upregulation of the stem-cell-like population in GBM. We first evaluated the differential expression of BC200 RNA in GBM (U87MG, DBTRG-05MG, T98G, and GBM8901) and normal human astrocytes (NHA) cells. BC200 RNA expression was significantly higher in GBM cell lines than in NHA cells (Figure 2A). The evaluation of cell viability after TMZ treatment revealed that more invasive glioblastoma cells GBM cells (GBM8901, T98G) showed higher chemo-resistance (TMZ) than the likely GBM cells DBTRG05MG and U87MG cancer cell (Figure 2B). Previous studies have suggested that the increased expression of ALDH+ and CD133+ cells was associated with mesenchymal phenotype and chemoresistance [19], cancer recurrence, and poor prognosis [20]. Furthermore, the role of BC200 RNA in TMZ resistance was assessed through flow cytometric analysis, revealing a significantly higher number of ALDH+/CD133+ cells in cancer cells than in normal human astrocytes (Figure 2C). The expression levels of key markers associated with drug resistance [21, 22, 23], proliferation, migration, and invasion, such as BCRP1, MDR1, MRP1 and MGMT along with BC200 RNA, were significantly highly expressed in GBM cells (Figure 2D).

Please also see the updated figure 2 legend, line 271-275.

Figure 2. Differential expression of BC200 RNA in GBM cell lines. (A) The differential expression of BC200 RNA in GBM cell lines and normal human astrocytes are shown. (B) The viability of GBM cells was analyzed through SRB assay 48 h after TMZ (0–1000 μM) treatment. (C) Flow cytometry analysis of the ALDH1+/CD133+ portion in GBM cell lines and normal human astrocytes. (D) The level of BC200, BCRP1, MDR1, MRP1 and MGMT in GBM cell lines was analyzed using qRT-PCR. *P < 0.05, **P < 0.01, and ***P < 0.001.

Reviewer 2 Report

Comment on the revised version by Su et al.:

Although new data were added, they were not considered in the discussion and the model presented in Fig. 8, and thus the model is still unclear. The legend in Fig. 8 states that ".....BC200 RNA enhances TMZ chemoresistance through modulating miR-2018-5p." This is not sufficient. Please go on, explaining the targets that are critically involved.

In the model, miR-2018 inhibits MGMT expresion, i.e. high BC200 results in low miR-218 and this results in high MGMT and, concomitantly, low toxicity driven by O6-methylguanine adducts that are repaired by MGMT. This hypothesis may apply to MGMT expressing cells, such as T98G, but not to U87MG and many other GBM cell lines  and 40% of tumors, in which MGMT is silenced by promoter methylation. Thus, the data obtained by U87MG and other MGMT lacking lines in this manuscript cannot be explained by this model.

This important point should be discussed and addresed in conclusions and the legend of Fig. 8.

Furthermore, new data on MMR were added, but not critically discussed as to its role in TMZ resistance. Please note that in MGMT silenced cells modulation of MMR is without effect on the TMZ-induced killing response. Please also note that ABC transporters are not involved in TMZ-induced responses, as suggested in Fig. 8.

Author Response

Point-by-point responses to reviewer’s comments:

We would like to thank all the reviewers for the thorough reading of our manuscript as well as their valuable comments. We have followed the reviewer’s comments thoroughly and feel that they have further helped in strengthening the manuscript.

Reviewer 2 Comments and Suggestions for Authors

  1. Although new data were added, they were not considered in the discussion and the model presented in Fig. 8, and thus the model is still unclear. The legend in Fig. 8 states that "BC200 RNA enhances TMZ chemoresistance through modulating miR-2018-5p." This is not sufficient. Please go on, explaining the targets that are critically involved. In the model, miR-2018 inhibits MGMT expresion, i.e. high BC200 results in low miR-218 and this results in high MGMT and, concomitantly, low toxicity driven by O6-methylguanine adducts that are repaired by MGMT. This hypothesis may apply to MGMT expressing cells, such as T98G, but not to U87MG and many other GBM cell lines and 40% of tumors, in which MGMT is silenced by promoter methylation. Thus, the data obtained by U87MG and other MGMT lacking lines in this manuscript cannot be explained by this model.

R1: We appreciate the reviewer’s insightful suggestion. We have made the change as requested by the reviewer. Please Kindly see our modified Fig 8 and also kindly see in Revised Discussion and Conclusion section, please see line 419-522.

  1. Discussion

Despite major developments in therapeutic interventions, GBM is one of the most lethal and invasive malignancies of the central nervous system and is the leading cause of overall cancer-associated mortality. Patients with GBM exhibit poor prognosis [26] and the current therapy mostly fails because these patients develop multidrug resistance and do not respond to the treatment [27, 28, 29]. Studies have shown that lncRNAs are widely accepted and play a crucial role in the development, progression, and metastasis of cancer as well as in drug resistance of GBM. The expression of BC200, a lncRNA, is high in the neuron-specific transcript in the brain [11]. The increased expression of BC200 RNA was atypically found in various human cancers, and its expression was higher in invasive cancers than in benign tumors [13, 14]. However, BC200 RNA as a clinical therapeutic target or biomarker for GBM is still understudied. In this study, the molecular mechanism of BC200 RNA in glioma development and progression was explored.

        Regulatory BC200 RNA is highly expressed in the peripheral blood of patients with breast cancer [30], suggesting that it can be an important molecular tumor biomarker for indicating human malignancies [31]. In our study, we demonstrated that BC200, both at protein and mRNA level, was significantly upregulated in blood and tissue samples of GBM patients (Figure 1). The expression of Ki-67 and p53-mut markers are indicative of prognosis and survival of GBM patients [17, 18], indicating a positive and significant correlation of BC200 RNA expression with Ki-67and p53. Thus, BC200 RNA is strongly related to poor prognosis of GBM patients.

        CSCs are now increasingly being recognized as a critical target in cancer treatment because these stem cells are often correlated with the resistance against conventional chemo- and radiotherapy. Overcoming their treatment resistance is the key issue in cancer therapeutics [32, 33]. Moreover, studies have reported that dysregulated lncRNA expression remains associated with the TMZ drug resistance of GBM. In this study, we observed that the expression of BC200 increased in GBM cells compared with normal brain cells (Figure 2). Furthermore, this study showed that GBM cells exhibited high expression of ALDH+/CD133+ cells, and key markers associated with drug resistance, proliferation, invasions and metastasis. TMZ resistance is a main reason for treatment fails. The causes of TMZ resistance are mainly DNA repair system, MMR is critical for inducing appropriate cellular responses to DNA damage, previous research suggests that GBM cells are TMZ sensitive when MMR is expressed and active [34]. Much of resistance to TMZ observed clinically is due to high expression of MGMT or loss of MMR [35, 36]. TMZ treatment induces DNA lesions and the MMR system causing apoptosis in GBM [37]. The combination of BC200 inhibition and TMZ treatment may lead to a new therapeutic strategy to improve the efficacy of TMZ in glioblastoma multiforme patients. ABC transporter such as BCRP1, MDR1, and MRP1, were associated with chemoresistance [9]. Moreover, we found that in GBM cells, the expression of these markers together with BC200 RNA expression is significantly higher, suggesting their role in TMZ chemoresistance. To further understand the role of BC200 RNA in GBM progression, we used the gain and loss of function of BC200 RNA through gene silencing and overexpression experiments. BC200 knockdown significantly reduced the proliferation, migration, and invasion of GBM cells and the expression of markers associated with the self-renewal and pluripotent ability by reducing the colony-forming and neurosphere generation (Figure 3). Meanwhile, BC200 overexpression significantly reduced the BC200 knockdown effect (Figure 4). Therefore, our result indicates that BC200 RNA plays a key role in GBM. This result is consistent with those of previous reports indicating the role of BC200 RNA as an oncogene in other cancers, such as breast, cervical, and colon cancers [38, 39].

Additionally, to understand the underlying molecular mechanism of BC200 in GBM chemosensitivity, a cell viability assay was performed. BC200-silenced cells showed decreased resistance to TMZ compared with the control group (Figure 5A), whereas OEBC200 showed the reverse effect (Figure 5). Western blot analysis showed a reduction in the expression of markers associated with drug resistance in BC200-knockdown cells, and the results were reverse in OEBC200 cells. A previous study indicated that miR in correlation with lncRNA (BC200) expression attenuated the viability, migration, and invasiveness of cancer cells [40]. Subsequently, bioinformatic analysis was performed to evaluate the potential miRNA expression in BC200-inhibited and -overexpressed GBM cells. Results from heatmap and binding prediction show that the expression of miR-218-5p is negatively associated with BC200 RNA expression, indicating that miR-218-5p targets BC200 RNA (Figure 5). Studies have shown that an increased expression of miR-218-5p affects cell viability in GBM cells [41], but the role of miR-218-5p in conferring chemosensitivity is still not studied. Our result confirms that miR-218-5p is an effective tumor suppressor miRNA that targets BC200 RNA, with high expression in normal cells than in GBM cells (Figure 6).

In our model, miR-2018 inhibits MGMT expression, i.e. high BC200 results in low miR-218 and this results in high MGMT and, concomitantly, low toxicity driven by O6-methylguanine adducts that are repaired by MGMT. Additionally, the miR-218-5p plays a key role in preventing the invasiveness of glioma cells. Zhixiao L et., al (2019) noticed that miR-218-5p can specifically bind to LHFPL3 mRNA and inhibit epithelial-mesenchymal transitions [42]. This is another possible reason to support these results. Furthermore, the relative expression of miR-218-5p was negatively correlated with BC200 RNA expression in GBM tissues. In vitro results showed that targeted inhibition of miR-218-5p significantly increased tumorsphere and colony-forming abilities in GBM cells compared with the control (NC) mock-transfected groups, confirming TMZ resistance. Again, we observed that markers associated with multidrug resistance, self-renewal, and pluripotency were upregulated in the miR-218-5p-inhibited group, (Figure 6) compared with the control mock-transfected groups, suggesting the role of miR-218-5p in GBM cell chemosensitivity, inhibit the expression of the MGMT and MMR system induces resensitization to TMZ. Although the relative expression of ABC-transport protein has regulated by BC200/mir-218-5p axis, but ABC-transport protein as inducers of the TMZ resistant phenotype in GBM is still controversial [43]. Finally, we demonstrated that the combination of BC200 inhibition and TMZ treatment increased GBM sensitivity. It greatly inhibited the tumor growth in the xenograft mouse model with no negative impact on body weight and survival (Figure 7). The results of IHC analysis are concordant with those of qRT-PCR; the shBC200/TMZ combination group shows significant suppression of tumor proliferation, oncogenicity, and tumorigenesis-associated markers. Moreover, the miR-218-5p level expressed was significantly high in the combination treatment group. Collectively, we provided a strong preclinical evidence in support of using shBC200/TMZ combination for treating malignant glioma cancer. In addition, miR-218-5p could be used as a biomarker for monitoring therapeutic responses.

  1. Conclusion

   TMZ resistance is one of the critical causes of treatment failure in GBM patients. The molecular mechanisms of TMZ resistance are still unclear. Hence, we used GBM cell lines exposed to TMZ treatment to analyze the relevance of MGMT and MMR system between BC200/miR-218-5p axis in the TMZ resistance phenomenon. In summary, as shown in schema abstract of Figure 8, our results suggested that BC200 RNA, a lncRNA, is highly expressed both in vitro and in vivo and significantly modulates GBM oncogenicity and enhances TMZ resistance through concomitantly enhancing self-renewal and pluripotency of GBM cells by modulating the expression of tumor-suppressor miR-218-5p. Our findings highlight the therapeutic efficacy of BC200 RNA as a clinical biomarker or therapeutic target for GBM.

The updated reference has added on the reference section, please see line 655-682.

  1. Sang Y.Lee. Temozolomide resistance in glioblastoma multiforme. Genes & Diseases 3(3), 198-210 (2016).
  2. Monika E Hegi. et al. Correlation of O6-methylguanine methyltransferase (MGMT) promoter methylation with clinical outcomes in glioblastoma and clinical strategies to modulate MGMT activity. J Clin Oncol 26(25), 4189-4199 (2008).
  3. Jann N Sarkaria. et al. Mechanisms of chemoresistance to alkylating agents in malignant glioma. Clin Cancer Res 14(10), 2900-2908 (2008).
  4. Koji Yoshimoto. et al. Complex DNA repair pathways as possible therapeutic targets to overcome temozolomide resistance in glioblastoma. Front Oncol 2(186), (2012).
  5. Wu, K. et al. Long noncoding RNA BC200 regulates cell growth and invasion in colon cancer. The international journal of biochemistry & cell biology 99, 219-225 (2018).
  6. Peng, J., Hou, F., Feng, J., Xu, S.X. & Meng, X.Y. Long non-coding RNA BCYRN1 promotes the proliferation and metastasis of cervical cancer via targeting microRNA-138 in vitro and in vivo. Oncology letters 15, 5809-5818 (2018).
  7. Li, Z. et al. Long non-coding RNA MALAT1 promotes proliferation and suppresses apoptosis of glioma cells through derepressing Rap1B by sponging miR-101. Journal of neuro-oncology 134, 19-28 (2017).
  8. Xia, H. et al. MiR-218 sensitizes glioma cells to apoptosis and inhibits tumorigenicity by regulating ECOP-mediated suppression of NF-kappaB activity. Neuro-oncology 15, 413-422 (2013).
  9. Zhixiao Li. et al. MiR-218-5p targets LHFPL3 to regulate proliferation, migration, and epithelial–mesenchymal transitions of human glioma cells. Biosci Rep 39 (3), (2019).
  10. Chiara Riganti. et al. Temozolomide downregulates P-glycoprotein expression in glioblastoma stem cells by interfering with the Wnt3a/glycogen synthase-3 kinase/β-catenin pathway. Neuro-oncology 15 (11), 1502–1517 (2013).

  1. This important point should be discussed and addresed in conclusions and the legend of Fig. 8. Furthermore, new data on MMR were added, but not critically discussed as to its role in TMZ resistance. Please note that in MGMT silenced cells modulation of MMR is without effect on the TMZ-induced killing response. Please also note that ABC transporters are not involved in TMZ-induced responses, as suggested in Fig. 8.

R2: Reviewer's comments are greatly appreciated. We appreciate the reviewer’s insightful suggestion. We have made the change as requested by the reviewer. Please Kindly see our modified Fig 8.

Please also see the updated figure 8 legend, line 533-535.

Figure 8. The lncRNA BC200 RNA is highly expressed both in vitro and in vivo and significantly modulates GBM oncogenicity through concomitantly enhancing self-renewal and pluripotency, enhances TMZ resistance through regulated MGMT and MMR system of GBM cells by modulating the expression of tumor-suppressor miR-218-5p.

Reviewer 3 Report

Most issues have been addressed. The following points still need to be clarified.

1. Spell out SRB when mentioned the first time.

2. Line 76, should be “also found”.

3. Product number of miR-218 in method should be provided. What is the difference between miR-218 and miR-218-5p?

4. Line 258, all four cell lines were included as GBM. Why discussion of Fig. 2B say GBM and “likely” GBM cells. This point has not been addressed.

5. Figs. 6E and F, inhibitor is spelt incorrectly.

6. Line 441, BC200 is lnRNA. Why mention its protein level?

Author Response

Point-by-point responses to reviewer’s comments:

We would like to thank all the reviewers for the thorough reading of our manuscript as well as their valuable comments. We have followed the reviewer’s comments thoroughly and feel that they have further helped in strengthening the manuscript.

Reviewer 3 Comments and Suggestions for Authors

Most issues have been addressed. The following points still need to be clarified.

  1. Spell out SRB when mentioned the first time.

R1: We appreciate the reviewer’s insightful suggestion. We have made the change as requested by the reviewer. Please kindly see line 83-86.

In this study, the role of BC200 RNA in GBM was evaluated to understand the molecular mechanism of chemoresistance and disease progression. The expression level of BC200 RNA in the tissue of GBM patients and GBM cell lines was evaluated. The SRB (Sulforhodamine B) assay was used to determine the effect of TMZ treatment on GB cells.

  1. Line 76 should be “also found”.

R2: We appreciate the reviewer’s insightful suggestion. We have made the change as requested by the reviewer. Please Kindly see line 76-77.

BC200 RNA was also found in blood specimens such as breast cancer and hepatocellular carcinoma [13,14].

  1. Product number of miR-218 in method should be provided. What is the difference between miR-218 and miR-218-5p?

R3: We appreciate the reviewer’s insightful suggestion. We have made the change as requested by the reviewer. Please Kindly see line 136-139.

2.5 Vector construction and infection

Lentivirus containing BC200 short hairpin (shBC200) RNA and BC200 overexpression (OEBC200) vectors were purchased from ThermoFisher Scientific (USA) and were used according to the manufacturer’s instructions. Two clones of shRNA were used to effectively knockdown (shBC200) and overexpress (OEBC200) BC200; the complete procedure of shRNA lentivirus infection and construction was conducted according to the practice guidelines at a certified BSL-2 laboratory, The Integrated Laboratories for Translational Medicine, Taipei Medical University. Human GB cells were transfected with miR-218-5p (mimic), miR-negative control (miR-NC), and miR-218-5p (inhibitor) were purchased from Qiagen and used as per the manufacturer’s protocol.

  1. Line 258, all four cell lines were included as GBM. Why discussion of Fig. 2B say GBM and “likely” GBM cells. This point has not been addressed.

R4: We appreciate the reviewer’s insightful suggestion. We have made the change as requested by the reviewer. Please Kindly see line 261-278.

3.2 High BC200 RNA expression associated with TMZ resistance of stem-cell-like population in GBM cells

We attempted to confirm the role of BC200 RNA expression in chemo-resistance through upregulation of the stem-cell-like population in GBM. We first evaluated the differential expression of BC200 RNA in GBM (U87MG, DBTRG-05MG, T98G, and GBM8901) and normal human astrocytes (NHA) cells. BC200 RNA expression was significantly higher in GBM cell lines than in NHA cells (Figure 2A). The evaluation of cell viability after TMZ treatment revealed that more invasive glioblastoma cells GBM cells (GBM8901, T98G) showed higher chemo-resistance (TMZ) than the likely GBM cells DBTRG05MG and U87MG cancer cell (Figure 2B). Previous studies have suggested that the increased expression of ALDH+ and CD133+ cells was associated with mesenchymal phenotype and chemoresistance [19], cancer recurrence, and poor prognosis [20]. Furthermore, the role of BC200 RNA in TMZ resistance was assessed through flow cytometric analysis, revealing a significantly higher number of ALDH+/CD133+ cells in cancer cells than in normal human astrocytes (Figure 2C). The expression levels of key markers associated with drug resistance [21, 22, 23], proliferation, migration, and invasion, such as BCRP1, MDR1, MRP1 and MGMT along with BC200 RNA, were significantly highly expressed in GBM cells (Figure 2D).

Please also see the updated figure 2 legend, line 271-275.

Figure 2. Differential expression of BC200 RNA in GBM cell lines. (A) The differential expression of BC200 RNA in GBM cell lines and normal human astrocytes are shown. (B) The viability of GBM cells was analyzed through SRB assay 48 h after TMZ (0–1000 μM) treatment. (C) Flow cytometry analysis of the ALDH1+/CD133+ portion in GBM cell lines and normal human astrocytes. (D) The level of BC200, BCRP1, MDR1, MRP1 and MGMT in GBM cell lines was analyzed using qRT-PCR. *P < 0.05, **P < 0.01, and ***P < 0.001.

  1. 6E and F, inhibitor is spelt incorrectly.

R5: We appreciate the reviewer’s insightful suggestion. We have made the change as requested by the reviewer. Please Kindly see updated Fig.6

  1. Line 441, BC200 is lnRNA. Why mention its protein level?

R6: We appreciate the reviewer’s insightful suggestion. We have made the change as requested by the reviewer. Please Kindly see line 441-448.

Regulatory BC200 RNA is highly expressed in the peripheral blood of patients with breast cancer [30], suggesting that it can be an important molecular tumor biomarker for indicating human malignancies [31]. In our study, we demonstrated that BC200, both at protein and mRNA level, was significantly upregulated in blood and tissue samples of GBM patients (Figure 1). The expression of Ki-67 and p53-mut markers are indicative of prognosis and survival of GBM patients [17, 18], indicating a positive and significant correlation of BC200 RNA expression with Ki-67and p53. Thus, BC200 RNA is strongly related to poor prognosis of GBM patients.
